# Pattern regulation in a regenerating jellyfish

Chiara Sinigaglia[†‡*], Sophie Peron[†], Jeanne Eichelbrenner, Sandra Chevalier, Julia Steger[§], Carine Barreau, Evelyn Houliston, Lucas Leclère[*]

Sorbonne Université, CNRS, Laboratoire de Biologie du Développement de Villefranche-sur-mer (LBDV), Villefranche-sur-mer, France

**\*For correspondence:**
chi.sinigaglia@gmail.com (CS);
lucas.leclere@obs-vlfr.fr (LL)

[†]These authors contributed
equally to this work

**Present address:** [‡]Institut de
Génomique Fonctionnelle de
Lyon (IGFL), École Normale
Supérieure de Lyon, CNRS UMR
5242, Lyon, France; [§]Department
for Molecular Evolution and
Development, Centre of
Organismal Systems Biology,
University of Vienna, Vienna,
Austria

**Competing interests:** The
authors declare that no
competing interests exist.

**Reviewing editor:** Phillip A
Newmark, Morgridge Institute
for Research, United States

**Abstract** Jellyfish, with their tetraradial symmetry, offer a novel paradigm for addressing patterning mechanisms during regeneration. Here we show that an interplay between mechanical forces, cell migration and proliferation allows jellyfish fragments to regain shape and functionality rapidly, notably by efficient restoration of the central feeding organ (manubrium). Fragmentation first triggers actomyosin-powered remodeling that restores body umbrella shape, causing radial smooth muscle fibers to converge around 'hubs' which serve as positional landmarks. Stabilization of these hubs, and associated expression of *Wnt6*, depends on the configuration of the adjoining muscle fiber 'spokes'. Stabilized hubs presage the site of the manubrium blastema, whose growth is Wnt/β-catenin dependent and fueled by both cell proliferation and long-range cell recruitment. Manubrium morphogenesis is modulated by its connections with the gastrovascular canal system. We conclude that body patterning in regenerating jellyfish emerges mainly from local interactions, triggered and directed by the remodeling process.

## Introduction

Regrowth of new structures and their integration into existing body parts during animal regeneration require varying contributions from cell proliferation and tissue remodeling, an issue already highlighted by early studies on planarians (*Morgan, 1901*). The (re)specification of positional information has been largely attributed to organizer-like centers and gradients of diffusible morphogens, modeled to pattern the surrounding cells in a concentration-dependent manner (e.g. [*French et al., 1976*; *Lander, 2013*; *Raz et al., 2017*; *Witchley et al., 2013*]). In planarians, *Hydra* polyps or vertebrate limbs, the site of injury transforms into a signaling center resetting positional information and triggering the regrowth of missing elements through proliferation and/or tissue remodeling (*Chera et al., 2009*; *Owlarn et al., 2017*). Alongside morphogen-based systems, structural and mechanical cues can also contribute to patterning. Supra-cellular actin fibers, for instance, were proposed to contribute mechanically to directing the orientation of body axis during regeneration of *Hydra* polyp fragments (*Livshits et al., 2017*).

Cnidarians, the sister group to the major animal clade Bilateria, display remarkable regenerative abilities. *Hydra* polyps, which are a classic regeneration model (*Galliot, 2012*), the colonial hydrozoan *Hydractinia* (*Bradshaw et al., 2015*; *DuBuc et al., 2020*) and the sea anemone *Nematostella vectensis* (*Amiel et al., 2015*; *DuBuc et al., 2014*; *Schaffer et al., 2016*) have become valuable systems for studying wound healing and whole-body regeneration. In contrast the jellyfish (medusa), a dispersive and sexually reproductive form, generated asexually from the polyp stage of numerous species belonging to the Medusozoa clade (*Leclère et al., 2016*), has received relatively little attention.

Compared to the polyp stage, medusae display complex tissue architectures, including striated muscles and well-defined organs. Jellyfish from the clade Hydrozoa show remarkable regenerative plasticity and thus offer an attractive system for studying both whole organism re-patterning and

organ restoration (*Hargitt, 1899*; *Hargitt, 1903*; *Morgan, 1899*; *Neppi, 1918*; *Schmid, 1974*; *Schmid and Alder, 1984*; *Schmid and Tardent, 1971*). Studies using wild-collected hydrozoan jellyfish, notably of the genus *Clytia*, concluded that a high proportion of individuals had undergone regeneration following damages caused by predation (*Mills, 1993*). Jellyfish from the clades Scyphozoa and Cubozoa also display some regenerative capacities (*Gamero-Mora et al., 2019*; *Stamatis et al., 2018*). A recent study showed that lost parts are not replaced in *Aurelia* juveniles following amputation, although rotational symmetry is restored by a muscle-powered process termed 'symmetrization' (*Abrams et al., 2015*).

The pioneering work of V. Schmid and P. Tardent provided an initial characterization of the regenerative capacity of hydrozoan medusae. Among other species, they documented the ability of wild-caught *Clytia hemisphaerica* to reconstitute organs and restore umbrella shape after diverse types of damage (*Schmid and Tardent, 1971*; *Schmid, 1974*; *Schmid et al., 1976*). *Clytia* is now a reliable laboratory model with extensive genomic and transcriptomic resources (*Houliston et al., 2010*; *Leclère et al., 2019*), amenable to gene function analyses (*Momose et al., 2018*). The complex life cycle, comprising two adult phases - the polyp and the medusa - can be completed reliably in the laboratory (*Houliston et al., 2010*; *Lechable et al., 2019*): the developing embryo becomes a ciliated larva, which settles and metamorphoses into a benthic polyp. The polyp propagates asexually, generating a colony. Specialized polyps generate swimming medusae, which develop gonads and ensure gamete dispersal. Polyp colonies can be kept for years, without reduction of reproductive potential, while medusae grow to adult size in two to three weeks and have a total lifespan of about two months. *Clytia* medusae show 4-fold rotational symmetry around the central feeding organ, termed the manubrium (*Figure 1A–D*). The distribution of elements within the medusa can thus be defined by positional values along a radial axis (i.e. from the umbrella center to the rim), and by their angular spacing (*Figure 1B*). *Clytia* jellyfish are made up of four identical morphological units, called quadrants. Each umbrellar quadrant harbors one radial gastrovascular canal, a gonad, one lobe of the tetraradially-organized manubrium (*Figure 1D*) and a segment of the circular peripheral canal connecting the tentacle bulbs (up to four per quadrant). The manubrium, tentacle bulbs and gonads harbor populations of stem cells (*Amiel and Houliston, 2009*; *Denker et al., 2008*; *Leclère et al., 2012*), have autonomous functions and can be regarded as true organs. The main component of the umbrella is an acellular connective layer, the mesoglea (*Figure 1C*), covered on its external surface, the 'exumbrella', by a simple monolayer of epithelial cells (*Kamran et al., 2017*). The concave face of the umbrella, termed 'subumbrella', is composed of three overlapping tissues layers (*Figure 1C*): (i) an inner epithelium associated with the mesoglea, (ii) an overlying cnidarian-specific tissue type comprising radially aligned smooth epitheliomuscular cells ('radial smooth muscles'), (iii) a ring of striated epitheliomuscular cells at the umbrella periphery which power swimming contractions, and are sandwiched between the other two layers (*Leclère and Röttinger, 2016*).

We employed cutting and grafting experiments to understand patterning principles during *Clytia* jellyfish regeneration and address the underlying cellular and molecular mechanisms. Distinguishable phases of wound healing, tissue remodeling and cell proliferation contribute, in a predictable manner, to the restoration of shape and missing organs. Actomyosin-driven wound constriction triggers the remodeling of umbrella tissues, causing the formation of a transient landmark, a 'hub' of radial smooth muscle fibers. Depending on surrounding tissue topology, the hubs will stabilize, predicting the site where a new manubrium will regenerate. *CheWnt6* provides a likely molecular link between the hub and manubrium formation, since contraction-dependent expression at the wound site is maintained only in stabilized hubs. Wnt/β-catenin signaling is essential for blastema onset and manubrium regeneration. We also show that cells mobilized from other organs (proliferating stem cells and digestive cells) fuel the manubrium anlage, and that connections to radial canals locally dictate the growing manubrium geometry. Pattern in regenerating *Clytia* medusae emerges thus from the integration of local interactions between structural elements. Based on our findings, we propose an actomyosin-based 'spoke and hub' patterning system to account for manubrium positioning, which translates the wound-induced remodeling process into a precisely located blastema, likely through an interplay with Wnt signaling.

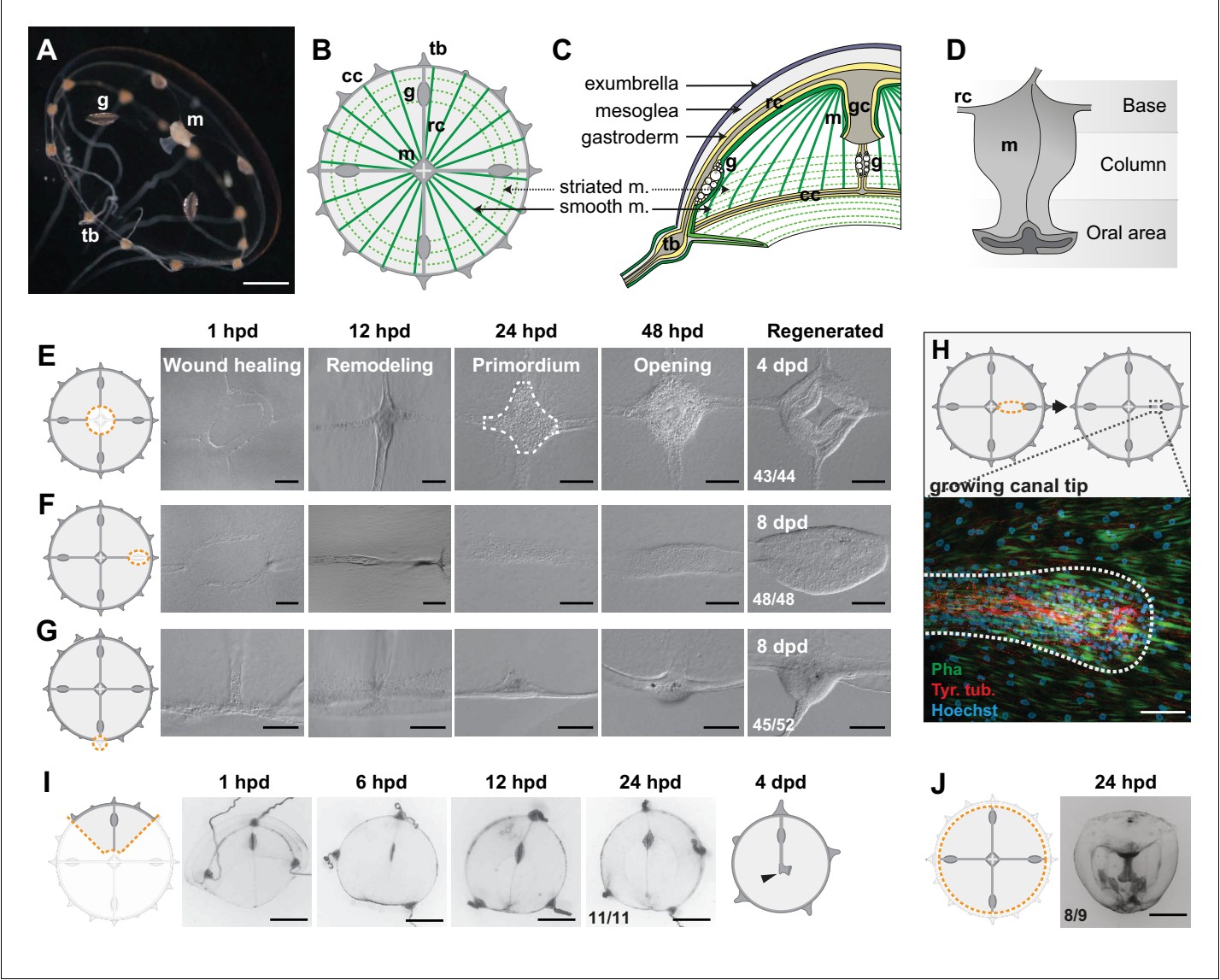

**Figure 1.** Regenerative potential of *Clytia hemisphaerica* jellyfish. (A–D) Anatomy of *Clytia* jellyfish. (A) Swimming female jellyfish (m: manubrium; g: gonads; tb: tentacle bulbs). (B) Tetraradial body organization: each umbrella quadrant comprises a radial canal (rc), a gonad (g), and up to 16 tentacle bulbs (tb). The tetraradially-shaped manubrium (m) lies at the center. A peripheral circular canal (cc) connects the tentacle bulbs. Radial smooth muscle fibers (smooth m.) and circular striated muscles (striated m.) line the subumbrellar surface. (C) The umbrella comprises an epithelial exumbrella layer, a connective mesoglea, and a subumbrella layer, constituted by an inner layer associated with the mesoglea and two layers of muscle fibers (smooth and striated). (D) Diagram of the manubrium, lateral view, showing the oral (distal)-aboral (proximal) axis. The base comprises four gastric pouches, connected to the four radial canals. The oral area is organized into four folds (lips). (E–J) Regenerative potential of *Clytia*. In each cartoon the cut is indicated by an orange dotted line, removed parts are veiled. (E) Manubrium regeneration is completed in 4 days, the white dashed line at 24 hpd circles the cell accumulation. (F) Gonad regeneration over 8 days. (G) Tentacle bulb regeneration over 8 days. (H) Detail of the tip of a regrowing radial canal shown by fluorescence microscopy; neural fibers associated with the canal are stained with an anti-tyr-Tub antibody (red). (I) A quarter jellyfish fragment that morphed within 24 hr into a small jellyfish. A tiny new manubrium was visible after 4 days (arrowhead). (J) Fragment from which the entire bell margin had been removed: it sealed upon itself without further regeneration. Abbreviations: (dpd) days post-dissection, (hpd) hours post-dissection. Scale bars- A,I-J: 1 mm; E-G: 100 μm; H: 20 μm.

The online version of this article includes the following source data and figure supplement(s) for figure 1:

**Figure supplement 1.** Organ regeneration in relation to feeding status.

**Figure supplement 1—source data 1.** Raw data for the plots shown in *Figure 1—figure supplement 1C–G*.

**Figure supplement 2.** Umbrella restorative potential.

## Results

### Restoration of medusa form involves both body remodeling and regrowth of organs

In order to gain insight into the self-organizing properties of the jellyfish *Clytia*, we explored its responses to a diverse array of dissections (*Figure 1E–J* and *Figure 1—figure supplement 1*, *Figure 1—figure supplement 2*).

Firstly, targeted ablations demonstrated that all organs (the feeding manubrium, gonads and tentacle bulbs) and radial canals can fully regenerate (*Figure 1E–H*). Functional manubria regenerated in 4 days (*Figure 1E*; n: 43/44), while gonads (bearing gametes) and tentacle bulbs reformed in about a week (*Figure 1F,G* and *Figure 1—figure supplement 1*; gonads: 48/48, bulbs: 45/52; after 8 days in fed jellyfish). Removed segments of radial canals rapidly reformed, re-growing from the stumps in both centripetal and centrifugal directions (*Figure 1H*; notice the growing tip). The recovery efficiency of gonads and tentacle bulbs varied, both within and between medusae (*Figure 1—figure supplement 1A–D*), while the manubrium recovered rapidly and stereotypically (*Figure 1—figure supplement 1E*). The efficiency of gonad and tentacle bulb regeneration was affected by the feeding regime of the animals, with recovery blocked at an initial stage of cell accumulation in starved animals (*Figure 1—figure supplement 1C,D*).

Secondly, a series of diverse cuts to the jellyfish umbrella allowed us to define the parameters of shape restoration (*Figure 1I,J* and *Figure 1—figure supplement 2*). Fragments began remodeling rapidly, reducing the wounded surface, and usually restoring the circular jellyfish shape within 24 hr, irrespective of the starting topology and of the number of remaining organs or canals (*Figure 1I* and *Figure 1—figure supplement 2*). This is well illustrated by half ('bisected') jellyfish or quarter fragments, which morph rapidly into a circular, smaller, umbrella (*Figure 1I* and *Figure 1—figure supplement 2C,D*). Together, the responses to our array of cut topologies indicated that: (i) existing organs and canals are largely conserved, (ii) the length of the residual circular canal determines the final perimeter of the restored umbrella, (iii) the final umbrella size correlates with the amount of tissue in the fragment (*Figure 1—figure supplement 2*). A rare failure in shape restoration concerned medusae from which the entire umbrella margin had been removed (*Figure 1J* and *Figure 1—figure supplement 2T*): these fragments sealed into a spherical shape and died shortly after. It is worth noticing that in the earlier study using larger, wild-caught *Clytia* jellyfish (*Schmid and Tardent, 1971*) reported one case of an interradial fragment, devoid of gastrovascular system, which regained a circular shape and an irregular manubrium. In cases where the manubrium was injured, we found that the organ rapidly healed before regaining its feeding function (*Figure 1—figure supplement 2C*). In manubrium-free fragments, cells accumulated at the injured ending of the remaining radial canal(s) within 24 hr post-dissection (hpd), and this mass regenerated into a new functional manubrium within 4 days post-dissection (dpd) (*Figures 1E* and *2A*). After remodeling and feeding, the umbrella of regenerated jellyfish continued to grow and new tentacle bulbs were added to the circular canal; however, if the original four-quadrant body plan had been lost, it never recovered. A minority of remodeled manubria sprouted additional radial canals (6/16 of bisected jellyfish showed a third radial canal after 6 weeks).

These observations highlighted the strong capacity for tissue and organ repair in *Clytia* jellyfish, but also showed that the final layout of body parts in regenerates does not necessarily match the original topology. Most obviously, missing radial canals were not restored, and the spacing between remaining canals and gonads was unbalanced in many cases (e.g. *Figure 1—figure supplement 2H, I,P*). Thus, restoration of the characteristic tetraradial symmetry of the jellyfish, manifest in the regular angular spacing of organs and canals (*Figure 1B*), requires at least part of all four radial canals (or manubrium corners from which they can sprout – see below) to be retained in the regenerate.

### Wound closure and body remodeling precede proliferation-dependent organ regeneration

We characterized the different processes of regeneration as follows: following manubrium ablation (*Figures 1E* and *2A*), the wounded exumbrellar and subumbrellar layers curl and fuse together, sealing off the exposed mesoglea (stage 0, *wound closure*). The hole in the umbrella starts to constrict, progressively pulling together the cut ends of the four radial canals. By 12 hpd the tissue gap is

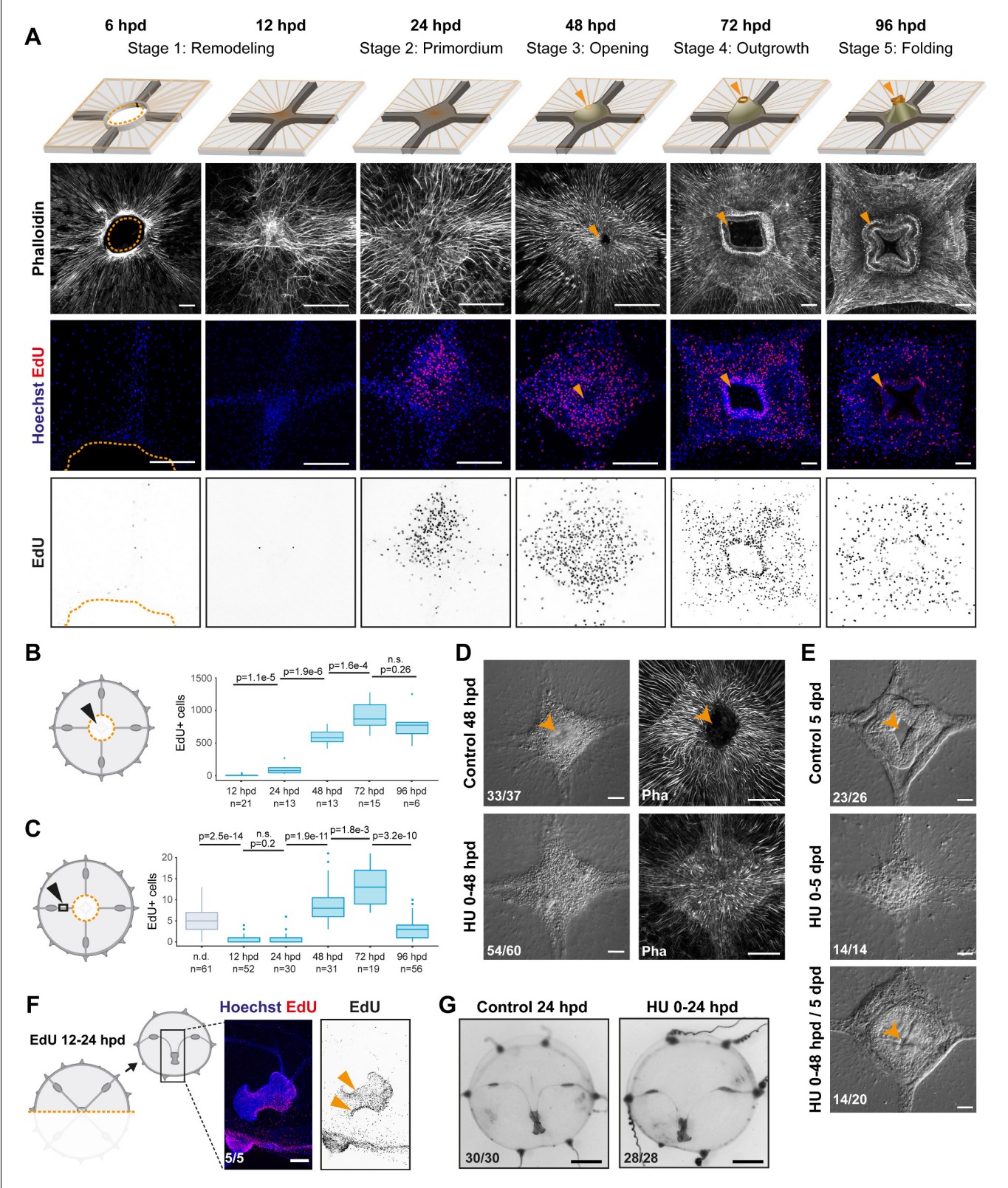

**Figure 2.** Manubrium regeneration, but not umbrella remodeling, requires cell proliferation. (A) Top: schematics of the different stages of manubrium regeneration following manubrium ablation. Below: Phalloidin staining (white) and EdU incorporation (1 hr pulse) at successive time points following manubrium ablation. EdU incorporation is detected from 24 hpd onward (quantification in B). Orange arrowhead: open gastric cavity from stage 3.
*Figure 2 continued on next page*

*Figure 2 continued*

Orange dotted line at 6 hpd: remodeling wound-area. The number of regenerating manubria used for quantification is indicated by the 'n' below each time point. (B, C) Quantification of EdU incorporation in the manubrium regeneration area (B) and in the radial canal (C). In (C), a 250 μm section of the MG-segment, at a standard distance of 250 μm from the manubrium, was considered for quantification. n.d.: radial canal from non-dissected medusae. Statistical test for (B) and (C): p values calculated with the Mann Whitney Wilcoxon test. Given that each time point is compared to the previous one, a Bonferroni correction was applied for identifying significant p-values: 0.0125 (B) and 0.01 (C), n.s. non-significant values. The number of individuals used for quantification is indicated by 'n' below each time point. (D) DIC (left) and phalloidin staining (right) images of regenerating manubria in control vs hydroxyurea (HU) treatment (0–48 hpd). HU treatment blocks blastema proliferation, but does not prevent the initial accumulation of cells. (E) DIC images of regenerating manubria, in control or HU-treated conditions (bottom: HU treatment for 0–48 hpd followed by washout). HU treatment prevents manubrium outgrowth and the correct formation of lobes. Orange arrowheads show the opening of the gastric cavity in control and HU-washed regenerating animals. (F, G) Remodeling does not require cell proliferation. (F) 24 hr after jellyfish bisection, EdU incorporation (12 hr pulse) is detected in the manubrium and close to the remodeling margin (left image: Hoechst-stained nuclei in blue, EdU in red; right: EdU in black). (G) Remodeled jellyfish fragments, in control conditions (left) or HU-treated (right). HU incubation does not affect shape restoration. Scale bars: A,D-F: 100 μm, G: 1 mm.

The online version of this article includes the following source data and figure supplement(s) for figure 2:

**Source data 1.** Raw data for the plots shown in *Figure 2B,C* and *Figure 2—figure supplement 1B,E*.
**Figure supplement 1.** Cell proliferation increases in the radial canals during manubrium regeneration, and is efficiently blocked by hydroxyurea treatment.

closed, and the radial canals meet at the center (stage 1, *remodeling*) and start fusing (n: 10/16 lumen fused at 12 hpd). Cells start accumulating at the junction of the radial canals, the first sign of organ regeneration; by 24 hpd the cell mass forms a roughly square, flat primordium (stage 2, *regeneration primordium*). The primordium thickens, and by 48 hpd it fissures centrally, revealing the now distinguishable gastric cavity (stage 3, *opening*). A thicker rim emerges, which develops into a short tubular outgrowth (stage 4, *outgrowth*). The protuberance elongates, and within 4 dpd, it develops the characteristic lip folds, tetraradially arranged; the manubrium is now fully functional, albeit smaller than the original one (stage 5, *folding*) (*Figure 1—figure supplement 1F,G*).

The contribution of cell proliferation to manubrium regrowth was assessed using one-hour incubations in the thymidine analogue EdU, which incorporates into newly-synthesized DNA (*Figure 2A–C* and *Figure 2—figure supplement 1*). Within the first hours following manubrium ablation, cell proliferation levels in the subumbrella decreased, notably within the segment of radial canal lying between gonad and manubrium (termed 'MG-segment'; *Figure 2A–C* and *Figure 2—figure supplement 1*). Cell proliferation increased at the regeneration site from 24 hpd, and within the radial canals from 48 hpd. Cell proliferation then peaked during primordium thickening stage (72 hpd), before returning to basal levels at 96 hpd (*Figure 2A-C* and *Figure 2—figure supplement 1A, B*). Treatment with the DNA synthesis inhibitor hydroxyurea blocked manubrium regeneration at a pre-opening stage (stages 2–3), thus establishing the dependency of manubrium outgrowth on cell proliferation (3 days n: 14/20; *Figure 2D, E* and *Figure 2—figure supplement 1D, E*). Conversely, hydroxyurea treatment did not impair wound healing, umbrella remodeling or the repositioning of regenerating manubrium after bisection (*Figure 2G*), consistent with the relatively low number of cells incorporating EdU detected during these processes (*Figure 2F*).

## Cell recruitment via the radial canals sustains manubrium morphogenesis

The high numbers of EdU-labelled cells detected within radial canals during the formation of the manubrium primordium (*Figure 2A, C* and *Figure 2—figure supplement 1A, B*) raised the possibility that precursor cells are recruited from other parts of the medusa through the canal system. Gonads and tentacle bulbs, which connect to the manubrium via the radial canals, harbor niches of multipotent stem cells (*Denker et al., 2008*; *Leclère et al., 2012*) called interstitial stem cells (i-cells; *Figure 3A* and *Figure 3—figure supplement 1A*). Hydrozoan i-cells can generate both somatic and germ cell types, and along with early precursors of these cell types, express stem cell markers such as *Nanos1* (*Bosch, 2009*).

In situ hybridization detection of *CheNanos1* in manubrium-ablated jellyfish revealed i-cells (used hereafter in a wide sense to include early precursors of the derivate cell types) in both regenerating manubrium and radial canals, starting from 24 hpd (*Figure 3B*). We could demonstrate that some of

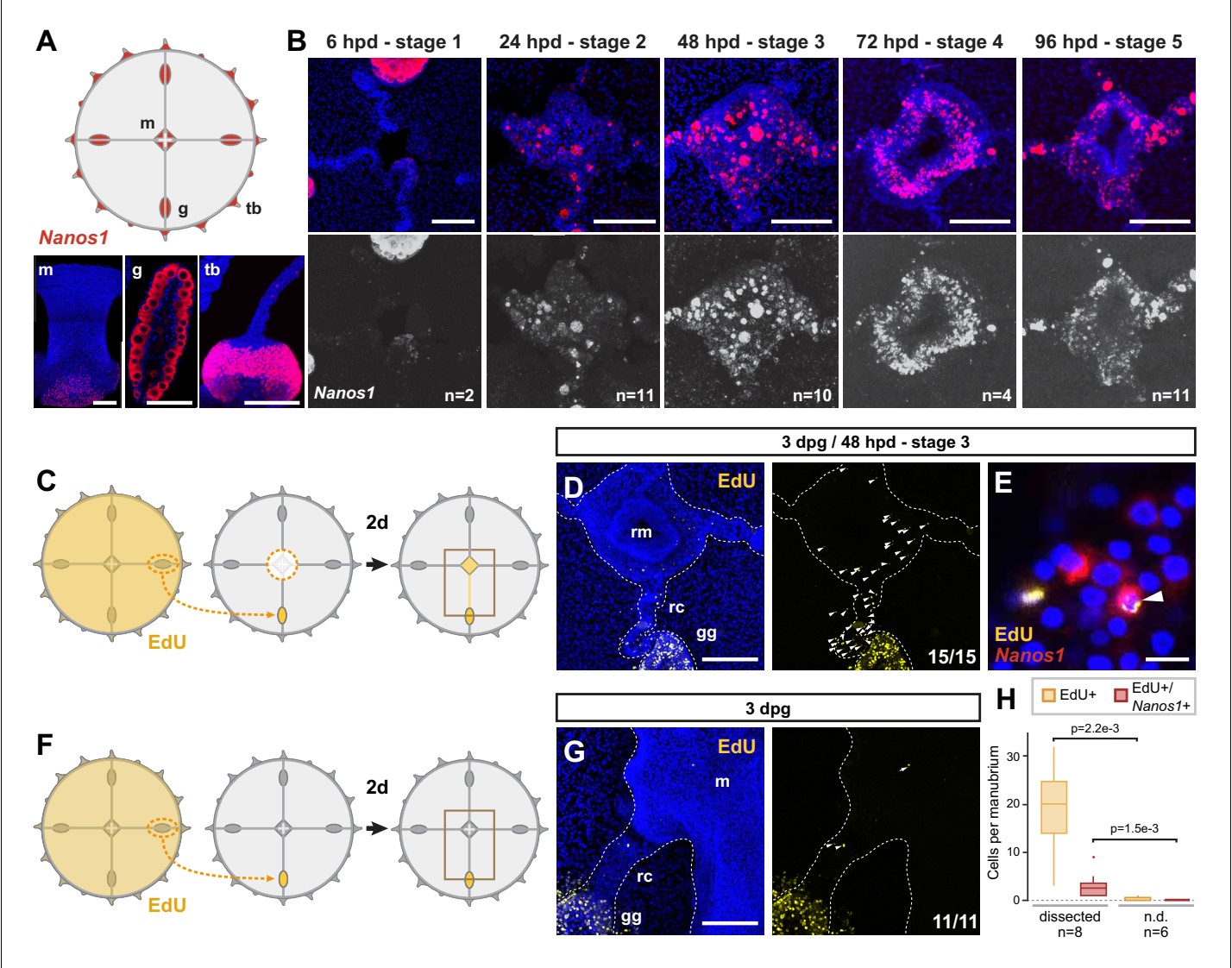

**Figure 3.** Dynamics of i-cell migration during manubrium regeneration. (A) Cartoon illustrates the distribution of *CheNanos1*-expressing cells (in red), also shown by FISH images in the bottom panels (*CheNanos1+* cells in red; nuclei counter-stained with Hoechst, in blue). *CheNanos1*-expressing cells reside at the base of the manubrium (m), in the gonads (g) and in a band of epidermis around the tentacle bulbs (tb). (B) FISH detection of *CheNanos1* in manubrium-ablated jellyfish, showing *CheNanos1*-expressing cells in the manubrium anlage and in the radial canals, at successive stages of manubrium regeneration. The larger cells are likely maturing oocytes. Colors in top row as in (A); *CheNanos1* mRNA detection alone is shown in white in the bottom images. (C–G) Grafting experiments detecting cell migration from gonad to the site of manubrium regeneration. (C) Diagram of experimental design: the donor medusa is incubated with EdU (24 hr pulse), one gonad is then excised and grafted at the place of one of the gonads of a non-treated host, which is then left to heal for 24 hr. Host manubrium is dissected, and regeneration proceeds for 48 hr. (D) Images from a host medusa 48 hpd - hours post manubrium dissection (3 days post graft - dpg). Left panel shows 2-channels image with EdU-positive cells (yellow), and nuclear counterstain (Hoechst, blue), right panel shows EdU-positive cells alone (white arrowheads). EdU-positive cells are detected within the radial canal connecting the grafted gonad to the regenerating manubrium, and in the manubrium primordium itself. (E) Higher magnification image from host medusa, showing putative migrating i-cell (white arrowhead), with colocalization of EdU and *CheNanos1* FISH staining - EdU cells (yellow), *CheNanos1* FISH (red), Hoechst nuclear counterstain (blue). (F) Diagram of control experiment, performed in parallel to (C): an EdU-labeled gonad (24 hr pulse) is grafted to a non-regenerating host (manubrium non dissected), let recover for 24 hr. EdU is detected in the host after two more days. (G) Images from a host control jellyfish; colors as in (D). Only few EdU-positive cells are detected in the non-dissected manubrium and the radial canal connected to the grafted gonad (white arrowheads; detail from larger image shown in *Figure 3—figure supplement 1G*). (H) Quantification of EdU-positive and EdU/ *CheNanos1* double positive cells in regenerating manubria compared to non-regenerating manubria; raw data: *Figure 3—source data 1*. Abbreviations: (m) manubrium, (rm) regenerating manubrium, (g) gonad, (gg) grafted gonad, (tb) tentacle bulb, (rc) radial canal, (dpg) days post grafting, (dpd) days post dissection. Statistical test: p values calculated with the Mann Whitney Wilcoxon test. Scale bars: A-D,G: 100 µm, E: 10 µm. The online version of this article includes the following source data and figure supplement(s) for figure 3:

*Figure 3 continued on next page*

*Figure 3 continued*

**Source data 1.** Raw data for the plots shown in *Figure 3H*.
**Figure supplement 1.** I-cell migration during manubrium regeneration.

these i-cells originated from the gonad by replacing a gonad in a host medusa prior to manubrium ablation with one from an EdU-labeled donor (24 hr incubation in EdU; *Figure 3C, D* and *Figure 3— figure supplement 1F-H*). Around 48 hr after manubrium ablation, EdU+ cells were detected in the host medusa both within the regenerating manubrium and in the connecting radial canal (*Figure 3D, H* and *Figure 3—figure supplement 1F-H*). A subpopulation of the migrating EdU+ cells expressed *CheNanos1* (*Figure 3E, H* and *Figure 3—figure supplement 1F-H*). EdU+/*CheNanos1*+ cells were found exclusively in the MG-segment of radial canals and in the manubrium primordium (*Figure 3— figure supplement 1F–I*), indicating that cell migration from the gonad is directed towards the regeneration site. Unexpectedly, small oocytes (significantly larger than the *CheNanos1*+ i-cells) labeled by EdU were also detected both in the radial canal and regenerating primordium (*Figure 3— figure supplement 1H*). The fate of these oocytes in the primordium is unclear. No mobilization of EdU+/*CheNanos1*+ cells from the gonad was observed in undamaged medusae (*Figure 3F–H*), suggesting that cellular recruitment is triggered by the regeneration process. The exact identity (early germ cells or somatic cell progenitors) and developmental potential (multipotent or restricted potential) of the smaller *CheNanos1*-expressing cells migrating from the gonad remains to be determined.

We noted another cell type, characterized by brown pigmentation, which circulated actively in the canals during regeneration and appeared to integrate into the manubrium primordium. In undamaged jellyfish, these cells were detected in the gastroderm of manubrium, gonads and tentacle bulbs (*Figure 4A*). These previously undescribed cells, which we term Mobilizing Gastro-Digestive (MGD) cells circulated rapidly within the lumen of the radial canal following manubrium ablation, likely transported by the flagellar beating of canal gastrodermal cells that govern nutrient flow (*Figure 4C*). We could label them specifically by feeding the medusa with fluorescent beads (*Figure 4A*, see Methods), supporting the idea that they function in nutrient uptake, and that the brown pigmentation derives from the crustacean diet (*Artemia* nauplii). We noted that the MGD cells also mobilized in the radial canals in starvation conditions, albeit less markedly than in regenerating jellyfish (*Figure 4D*), suggesting a role in adjusting nutritional balance between organs. Analysis of bead-labeled jellyfish following manubrium dissection revealed that some of the gastrodermal cells of the regenerating manubrium did indeed originate from other organs of the jellyfish (*Figure 4B*). Transplantation of a gonad from a medusa fed with fluorescent beads to an unlabeled host prior to manubrium ablation confirmed that MGD cells from the gonad gastroderm are recruited to the primordium of the regenerating manubrium, as well as to other non-regenerating bulbs and non-dissected manubria (*Figure 4E*). In medusae from which all gonads had been removed, fluorescent bead-labelled MGD cells were detected in the manubrium primordium once the radial canals had regrown, showing that these cells can be recruited not only from gonads but also from more peripheral regions, likely from the gastroderm of the tentacle bulbs (*Figure 4F*). On the other hand, no fluorescent bead-labelled MGD cells were found in the manubrium primordium if both gonads and radial canals were completely ablated (*Figure 4F*). This demonstrates that MGD cells travel uniquely through the gastrodermal canal system.

Related experiments demonstrated that intact radial canals are necessary for manubrium regeneration. The removal of all four radial canals (including gonads), leaving just their most proximal segments intact, led either to stalling of regeneration at early stages (stages 2–3, open cavity) or to formation of abnormal manubria with incorrect geometry and disrupted morphogenesis, notably lacking the deep folds shaping the four edges of the manubrium column (*Figures 4F* and *5A*). Correct morphogenesis of a manubrium only occurred when at least one radial canal was connected to the peripheral gastrovascular system (*Figure 5B*), with the details of its morphology conditioned by the number of canals (see below).

Based on these results, we propose that radial canals provide a route for the recruitment of diverse cell types, whose respective contributions to the regeneration of manubrium remain to be fully investigated. We observed mobilization of two cell types with distinct patterns of behavior: stem cells specifically migrate towards the regeneration site, while digestive cells circulate through

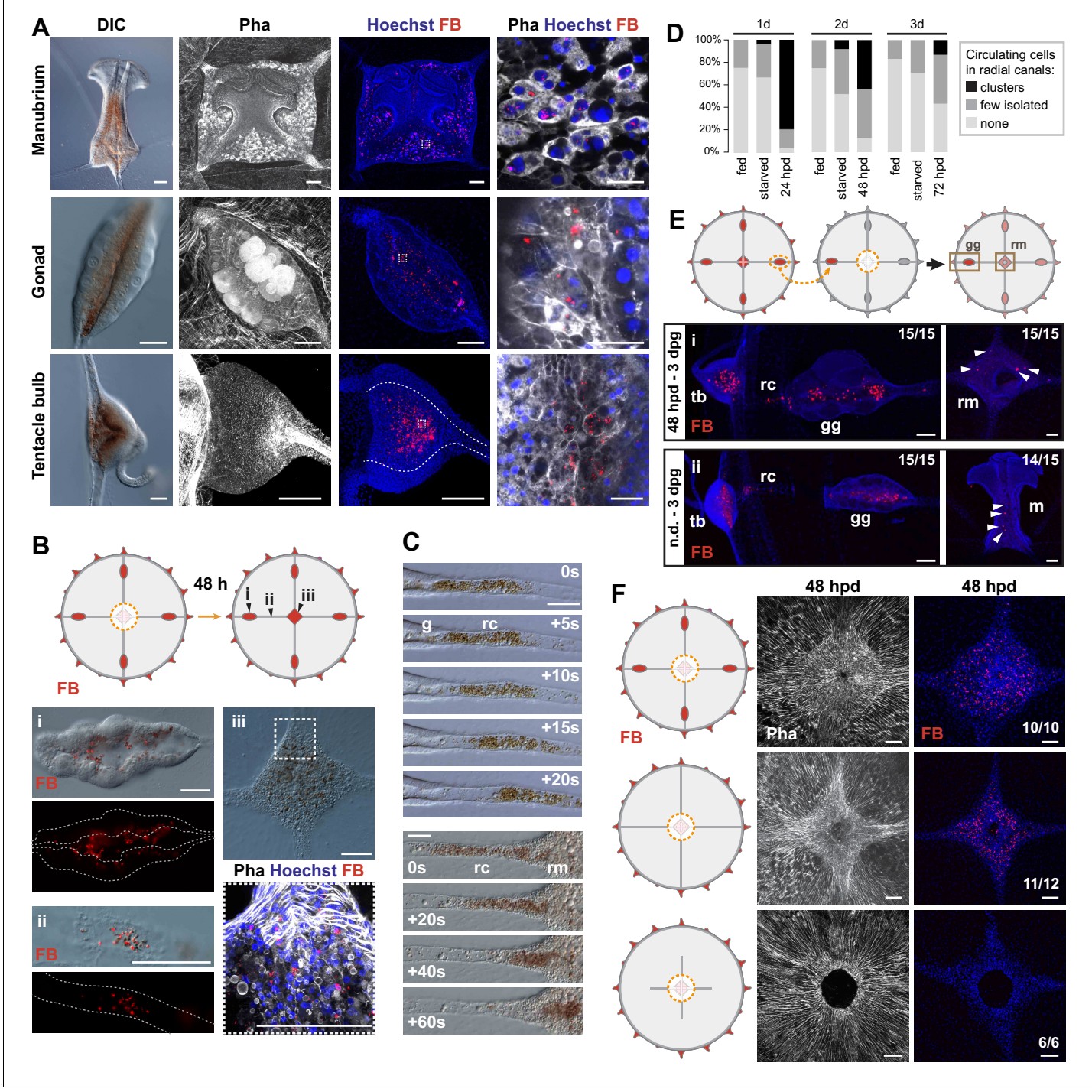

**Figure 4.** Non-targeted cell recruitment of Mobilizing Gastro-Digestive (MGD) cells during regeneration. (A) Images illustrating the distribution of MGD cells in *Clytia* organs (manubrium, gonads, tentacle bulbs), as indicated for each row. MGD cells can be recognized by the orange/brown pigment (DIC images in left panels). Second and third panels: Fluorescent micro-Beads (FB, in red) are ingested by cells residing in the gastroderm. Nuclei are counterstained with Hoechst (in blue), the organization of tissues in shown by phalloidin staining (in white). Right panel: higher magnification images including all three channels showing MGD cell morphology. All confocal images are maximum projection except the close-up image of the manubrium MGD cells that is a single plane of a z-stack. (B) Tracking of FB-labeled cells (following 24 hr feeding) in manubrium-regenerating medusae, cartoon illustrates experimental design. Distribution of FB-labeled MGD cells, 48 hr after manubrium dissection: DIC and fluorescence images (colors as in A) show FB-labeled cells in the gonad digestive cavity (*i*), in the radial canals (*ii*) and in the regenerating manubrium (*iii*). (C) Frames from time lapse movies of the MG-segment of radial canal (rc), showing rapid movement of MGD cells in the lumen. The five top panels show outward movement from the gonad (g), the bottom series shows movement into the regenerating manubrium (rm). (D) Quantification of circulating cells in the MG-segment of all

*Figure 4 continued on next page*

*Figure 4 continued*

four radial canals in fed, starved and regenerating jellyfish. Starved and dissected animals were last fed 8 hr before experiment (n: 24 for each condition, see *Figure 4—source data 1*) (**E**) Grafting of a FB-labeled gonad to a naïve host (cartoon illustrates experimental design) demonstrates untargeted migration of FB-labeled cells into all organs, either in regenerating jellyfish (manubrium dissected, *i* - top panel) or in non-dissected condition (n.d. *ii* – bottom panel). Panels show, from left to right: tentacle bulb, gonad, manubrium; FB in red, nuclear counterstain (Hoechst) in blue. (**F**) Experiment to determine the source of FB-labeled cells during manubrium regeneration (48 hpd), cartoons illustrate the three experimental conditions (from top to bottom): manubrium-dissected jellyfish, manubrium- and gonad-dissected (connection to tentacle bulbs is maintained), manubrium-dissected and canal-disconnected (gonads removed). Images show phalloidin staining (in white, to the left) visualizing the tissue organization around the manubrium site and fluorescent micro-beads (FB, right column, shown in red with Hoechst staining of nuclei in blue). FB-labeled cells are present in the regenerating manubrium in the first two conditions, but if there is no connection to another organ, manubrium regeneration is stalled, and no FB-labeled cells can be detected. Scale bars: 100 µm, except in the right column of (**A**): 25 µm.

The online version of this article includes the following source data for figure 4:

**Source data 1.** Raw data for the plots shown in *Figure 4D*.

the canal system and can settle in the gastroderm of any organ, including the manubrium regeneration site.

## Radial canals dictate the geometry of the regenerating manubrium

The tetraradial symmetry of the native manubrium mirrors the global symmetry of undamaged medusae, and is perfectly restored in regenerating manubria, if they are properly connected to the canal system (see above; *Figure 5J*). In order to explore further the link between manubrium symmetry and the global tetraradial symmetry of the jellyfish, we examined the geometry of manubria regenerated in the context of different cut topologies.

Bisected and quarter medusa fragments regenerated double- and single-lobed manubria, respectively (*Figure 5C,D*). Furthermore, irrespective of the dissection topology, the number of lobes of the regenerating manubrium always reflected the number of remaining radial canals, such that three radial canals generated a three-lobed manubrium, and so on (*Figure 5C–F*). This pattern might depend either on an underlying system of circular positional information retained within the medusa fragments, or by signals coming from the canals themselves. To distinguish these possibilities, we systematically removed radial canals from manubrium-ablated (but otherwise undamaged) medusae. The presence of four canals led in most cases to a tetraradial manubrium, three canals to a three-lobed one, two canals to a two-lobed manubrium, and one canal to a tubular one (*Figure 5G–J*). We conclude that the topology of the regenerating manubrium is determined by the number of connecting canals, which might provide a mechanical or biochemical signal, and/or might materially contribute to the primordium composition by providing migrating cells (*Figures 3* and *4*). Arguing against the latter option, canal fragments connected to the primordium but not to any other organ were found to be sufficient to direct the geometry of the regenerated manubrium (*Figure 5B*), as long as at least one canal remained connected to other organs (gonads and/or bulbs). Supporting our findings, *Neppi, 1918* reported a correlation between the number of canals and manubrium morphology in regenerating wild-caught *Clytia* jellyfish, without specifically testing this relationship. Canals appear therefore to direct morphogenesis of the proliferating primordium by a local influence, whose nature remains to be determined.

## Muscle fiber 'hubs' predict the site of manubrium regeneration

Recovering medusa fragments usually regenerate only a single manubrium. A key issue is thus to understand how the position of a new manubrium is determined. In undamaged jellyfish, the manubrium is located at the geometrical center of the umbrella, at the point of convergence ('hub') of the radial smooth muscles fibers and of the four radial canals. In regenerates, the initial position of the manubrium anlage does not necessarily coincide with the geometrical center of the remodeled medusa (e.g. in quarter jellyfish fragments; *Figure 6A* and *Figure 6—figure supplement 1A*). The anlage was, however, invariably associated with radial canal(s) and an aggregate ('hub') of smooth muscle fibers. We thus explored the contribution of the canals and/or the smooth muscle fibers hub to defining the position of the new organ.

Experiments presented above showed that complete manubrium regeneration is dependent on the presence of radial canals. Cells recruited through the radial canals are necessary for manubrium

growth and morphogenesis (see above, *Figure 5*). Consistently, interradial fragments lacking a radial canal showed no signs of primordium formation at 4 dpd (*Figure 6B*). After two weeks, however, a simple tube-like manubrium (unable to feed) was observed in a few cases (n: 13/65; *Figure 6B*). This observation indicates that the canals are not strictly required to position the new manubrium and that umbrella tissues also contribute to the new structure. Canals rather play a facilitating role to manubrium regeneration by contributing with specific cell types to the blastema. We can further conclude that the meeting of multiple canal ends (often seen during remodeling, see for example *Figure 1—figure supplement 2*) is not responsible for specifying the position of manubrium regeneration, as evidenced by quarter jellyfish fragments in which a manubrium regenerated on the sole surviving radial canal (*Figure 6A*).

Strikingly, F-actin staining showed that primordia and regenerated manubria were systematically associated with the aggregate (hub) of smooth muscle fibers that formed during the remodeling process (*Figure 6A*; n: 22/22). In all cases, a muscle hub could be detected before the anlage formed (*Figure 6A* and *Figure 6—figure supplement 1B*), irrespective of the wound topology. Hub formation did not require radial canals, since it was unaffected by removal of the MG-segment of radial canals after manubrium excision (*Figure 6C*). In those canal-ablated medusae, manubrium regeneration commenced when at least one of the excised canals regrew to the hub of smooth muscle fibers (*Figure 6C*; primordium on hub at 4 dpd: 8/9; manubrium at the center of umbrella at 5 dpd: 18/18). Taken together, these findings suggest that the position of the hub of smooth muscle

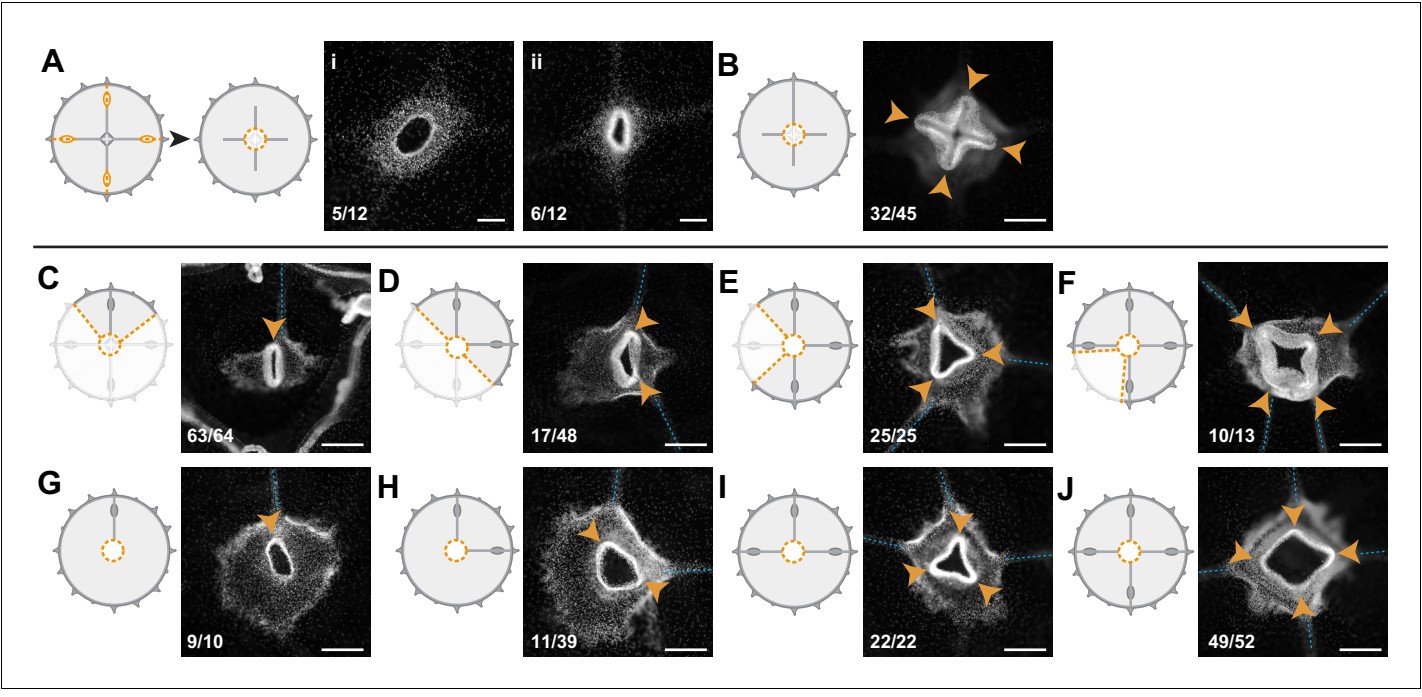

**Figure 5.** Radial canals locally dictate the morphology of regenerating manubrium. Cartoons illustrate experimental designs, with orange dotted lines showing cuts; fluorescence images show nuclear Hoechst staining (in white) of regenerated manubria (6 dpd), with blue dashed lines and orange arrowheads highlighting canals and lobes of the regenerated manubrium, respectively. Quantification of phenotypes at the bottom left of each image. (A) Experiment testing the contributions of gonads and tentacle bulbs to manubrium morphogenesis. All four gonads and the peripheral portions of the four radial canals were removed, prior to manubrium ablation. Two main morphologies were obtained (6 dpd): (*i*) arrested manubrium regeneration; (*ii*) impaired manubrium regeneration. (B) Similar to (A), but one complete canal (connected to another organ) has regrown prior to manubrium regeneration: normal morphogenesis occurred. Orange arrowheads point to the regenerated manubrium lobes. (C–J) Experiments demonstrating the correlation between the number of connected radial canals and regenerating manubrium morphology. Cartoons illustrate the strategy for manipulating number of canals: removal of either entire umbrella quadrants (in C-F), or of the gonad/canal system (in G-J). (C, G) Tubular regenerates formed when only a single canal is connected. (D, H) Bi-lobed regenerates formed in the presence of two connecting canals, the two lobes (orange arrowheads) follow the angular orientation of canals. (E, I) Three-lobed regenerates, connected to three radial canals. (F, J) Tetraradial manubria regenerated in the presence of four connecting canals. Scale bars: 100 μm.

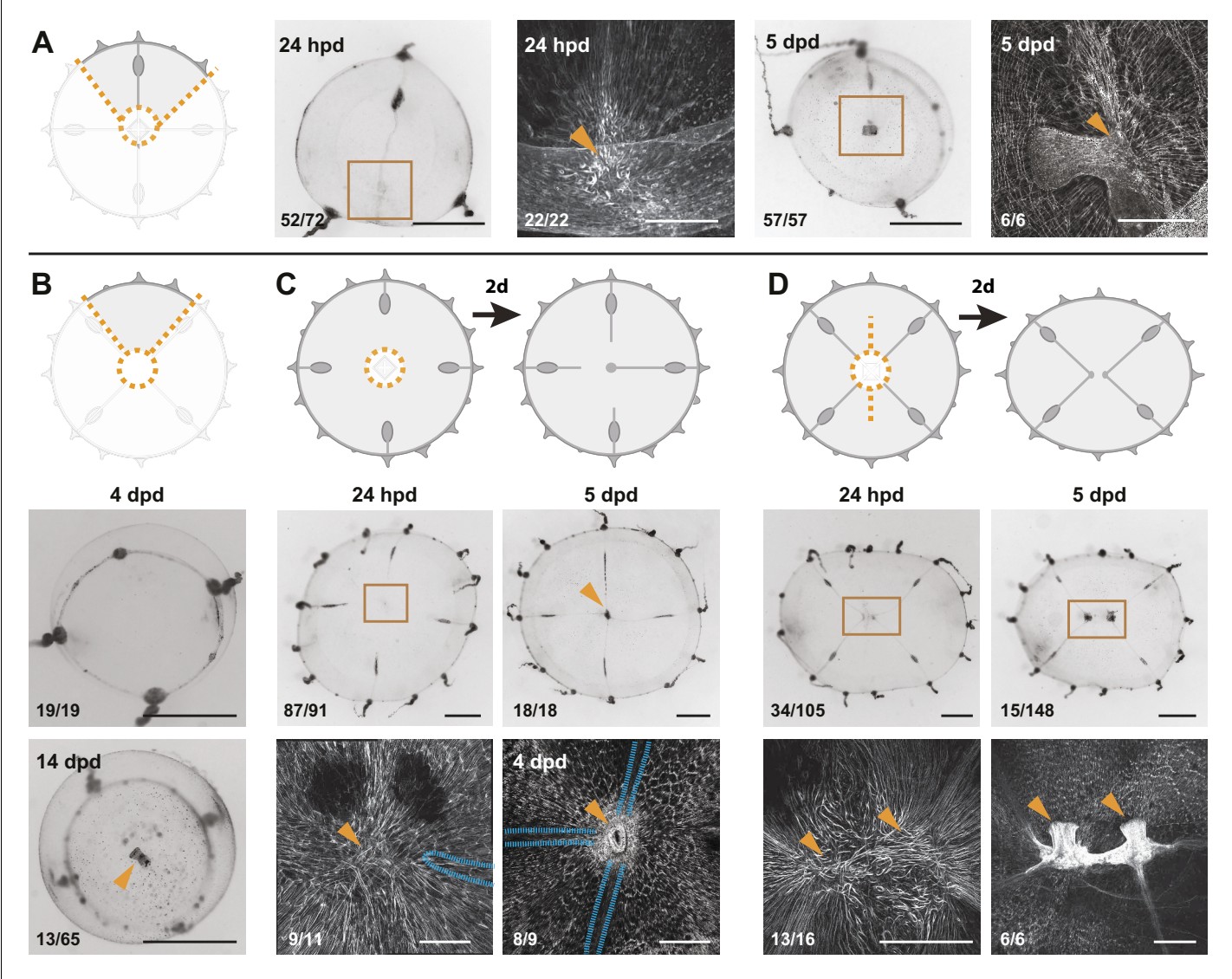

**Figure 6.** Smooth muscle hubs mark the position of regenerating manubria. For each experiment, cartoons illustrate the cut (orange dotted lines), and stereomicroscope images the responses of live regenerates, while fluorescent images show phalloidin staining (in white) of the regions outlined by brown squares in the stereomicroscope images. Quantification of phenotypes at the bottom left of each image. (**A**) Quarter fragments: 24 hpd, a radially arranged hub of muscle fibers (orange arrowhead) is detected away from the wound site, close to the injured end of the remaining radial canal. At 5 dpd, a regenerated manubrium is located at the smooth muscle hub. (**B**) Umbrella fragments devoid of radial canals observed after 4 dpd (top panel: re-circularized, no manubrium regeneration) and 14 dpd (bottom panel: a tubular manubrium-like structure, unable to feed, rarely forms). (**C**) Removal of canal segments, between the ablated manubrium and gonads: at 24 hpd (left) a muscle hub - but no manubrium blastema - is visible; regrowing radial canals (blue outline in phalloidin image) have not yet reached the hub. At 5 dpd (right panel) three radial canals (blue dashed line) are connected to the muscle hub and manubrium regeneration has occurred (orange arrowhead). (**D**) Manubrium dissection, coupled to a long cut through the umbrella: 24 hpd (left panel) two muscle hubs are visible (orange arrowheads). At 5 dpd (right) two manubria have formed (orange arrowheads), at the corresponding locations. Scale bars: A-C: 150 μm; D: 300 μm; for all stereoscope images: 1 mm.

The online version of this article includes the following source data and figure supplement(s) for figure 6:

**Figure supplement 1.** Multiple muscle hubs correlate with the regeneration of multiple manubria.
**Figure supplement 1—source data 1.** Raw data for the plots shown in *Figure 6—figure supplement 1D*.

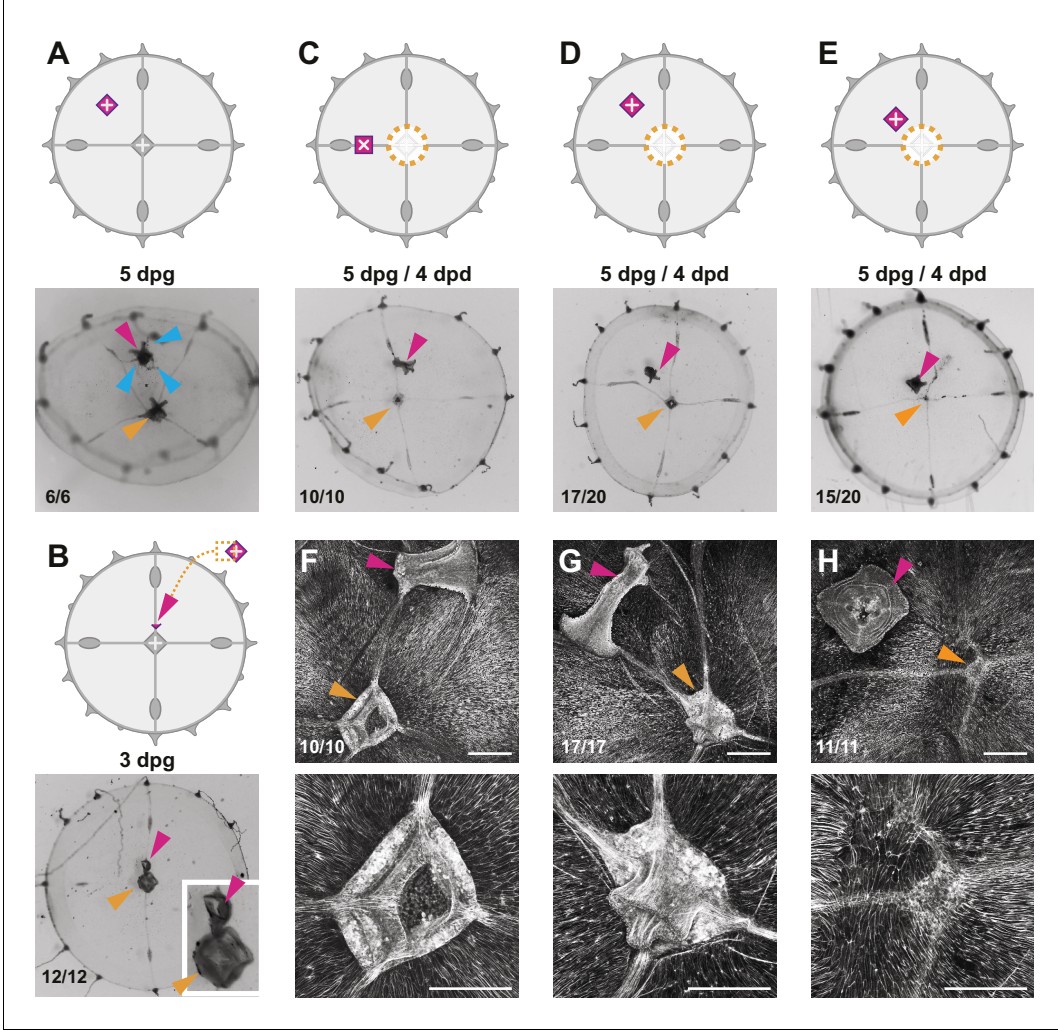

**Figure 7.** Effects of additional manubria on manubrium regeneration. Cartoons illustrate the grafting experiments, showing the ablation (orange dotted lines) and the grafted manubria position (in magenta). In (A–E), stereomicroscope images show live regenerates, in (F–H) fluorescent images of phalloidin staining (in white). Quantification of phenotypes at the bottom left of each image. (A) Grafting of an additional manubrium on the umbrella of a non-dissected jellyfish. After 5 days, the jellyfish bears two manubria (magenta and orange arrowheads indicate the grafted and the endogenous one, respectively), and the grafted manubrium has sprouted radial canals (blue arrowheads) from its base. Canals have no apparent directionality, and connect to the endogenous canal system. (B) Grafting of a fragment of manubrium base onto a non-dissected medusa, on a radial canal. After 3 days an extra manubrium is visible (magenta arrowhead). (C–E) Effect of ectopic manubria on manubrium regeneration. Grafting was performed one day prior to manubrium ablation. When manubria were grafted onto the radial canal (C) or in the middle of the umbrella (D), the endogenous manubria regenerated (orange arrowheads). (E) When the additional manubrium was grafted adjacent to the regeneration site, no manubrium regenerated (orange arrowhead). (F–H) Phalloidin staining of specimens corresponding to (C–E), showing graft-dependent muscle fiber topology: in manubrium-dissected jellyfish, muscle hubs form when the grafted manubrium was positioned on a radial canal (F) or on the umbrella (G) (orange arrowheads in the lower magnification images of the upper row), while muscles are disorganized when the graft is close to the ablation site (regeneration in this case is inhibited; orange arrowhead). (H) The radial canals have reconnected at the ablation site in all cases. (dpg) days post grafting. Scale bars: 100 μm.

fibers plays a key role in defining the location of the new organ, while connection to the radial canals fuels primordium growth and morphogenesis.

In order to test the relationship between the muscle fibers hub and the site of manubrium regeneration, we performed a longitudinal deep cut through the umbrella of manubrium-ablated medusae

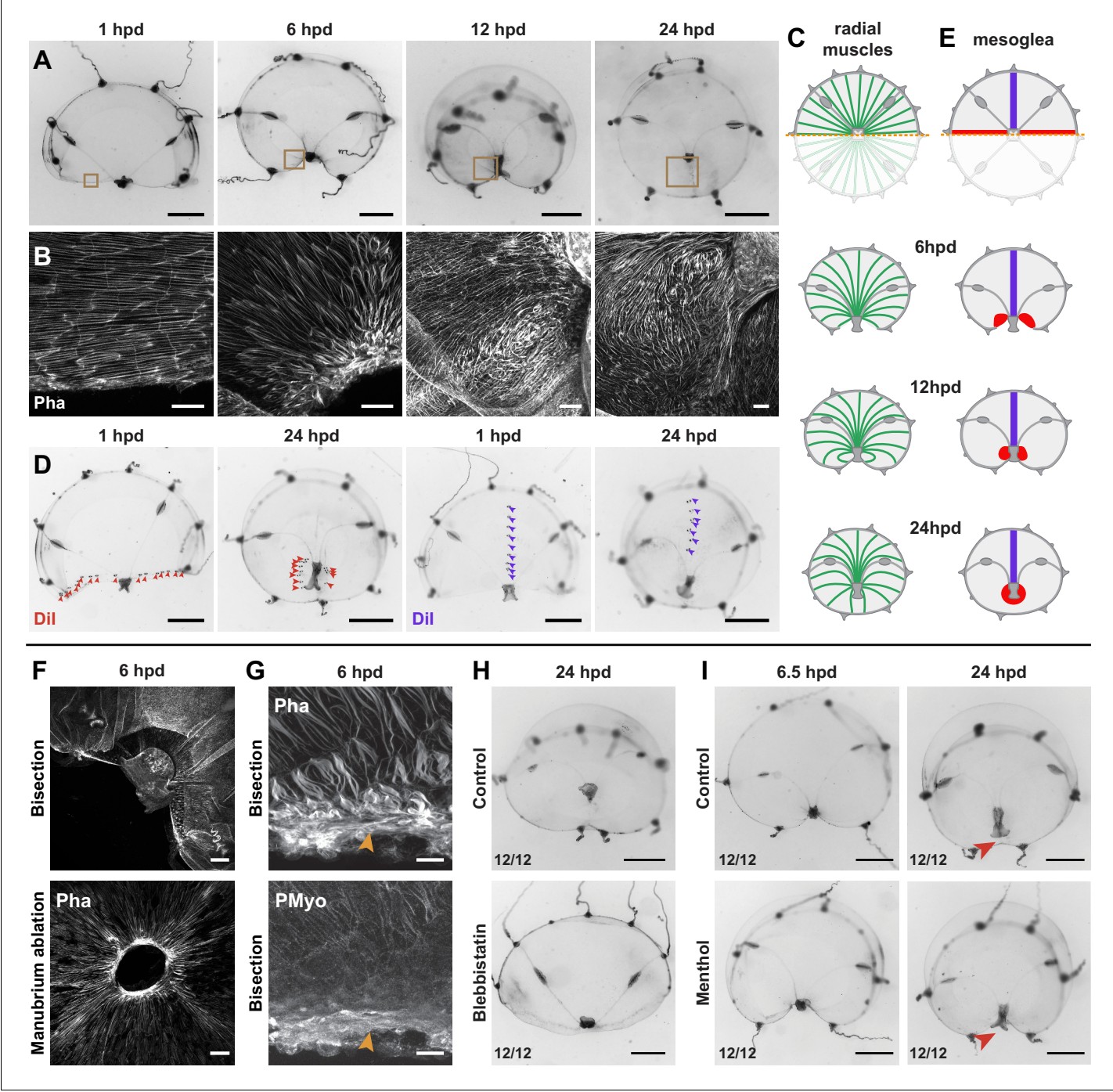

**Figure 8.** Actomyosin-driven subumbrella remodeling restores medusa shape. (**A**) Morphology of remodeling bisected medusae; successive stereomicroscope images showing the rapid reduction of exposed edges. After closure of the wound, the manubrium migrates towards the center, by 24 hpd only a scar-like trace is visible in the umbrella, which will then disappear. (**B, C**) Muscle fiber organization in remodeling jellyfish: in (**B**) phalloidin staining of the remodeling edge (corresponding to the regions highlighted by brown squares in A), in (**C**) diagram. The initially ordered radial fibers disorganize at the wound edge, and are extensively rearranged following the movements of the umbrella. (**D, E**) Images showing DiI droplets (in D) injected into the thick mesoglea layer of hemi-medusae, along the wound (left images, DiI droplets shown with red arrowheads, n: 8/8 jellyfish) or perpendicular to it (right images, DiI droplets shown with purple arrowheads, n: 6/6 jellyfish). Mesoglea was rearranged close to the wound area, while its more distal region seemed unaffected (diagram in E). (**F**) Phalloidin staining showing an actomyosin cable assembled at the wound edge in bisected (top) and manubrium-dissected (bottom) jellyfish. (**G**) Phalloidin (top) and antibody staining (PMyo; bottom) recognizing phospho-Myosin Light Chain 2 (Ser19) of the same specimen (6 hpd) showing enrichment of both actin and phosphorylated myosin at the wound site (orange arrowheads), during remodeling. (**H**) Halved jellyfish incubated (bottom) or not (top) with the myosin inhibitor blebbistatin: remodeling has not occurred in the presence of

*Figure 8 continued on next page*

*Figure 8 continued*

blebbistatin. (I) Halved jellyfish incubated (bottom) or not (top) in menthol, which inhibits striated muscle-based rhythmic contractions (see *Figure 8—figure supplement 1C*). Wound closure and remodeling (red arrowhead) have occurred in the menthol-treated specimen, with only a slight delay. Scale bars: A,D: 1 mm; F: 50 µm, B,G: 25 µm, H, I: 1 mm.

The online version of this article includes the following source data and figure supplement(s) for figure 8:

**Figure supplement 1.** Actomyosin-driven subumbrella remodeling restores medusa shape.

**Figure supplement 1—source data 1.** Raw data for the plots shown in *Figure 8—figure supplement 1C*.

(*Figure 6D* and *Figure 6—figure supplement 1D*). This cut topology disrupted the wound closure and remodeling processes, resulting in the formation of two separate manubria primordia (*Figure 6D*; n: 34/105, at 24 hpd) located on two muscle hubs (n: 13/16). In about a third of cases these regenerated two independent manubria, sometimes linked by a radial canal (*Figure 6D*). In other cases, the two manubria anlagens fused into a single oral structure, correctly patterned or twinned (*Figure 6—figure supplement 1D*; fused manubria, 5 dpd: 24/159). These 'double hub' experiments confirmed that (i) the muscle hub is a product of the remodeling process and its formation is independent of the radial canals (*Figure 6C* and *Figure 6—figure supplement 1C*), and (ii) each hub marks the location of a regenerating manubrium, irrespective of the subsequent outcome (two manubria or fused; *Figure 6D* and *Figure 6—figure supplement 1D*).

## Why does only one manubrium usually regenerate?

Our initial survey showed that any medusa fragment can reform a manubrium. This implies that the entire radial axis of the umbrella possesses the potential to form a manubrium, and that a control mechanism exists to prevent multiple manubria from forming. One explanation could be that *Clytia* manubrium has an organizing role, as is the case for the hypostome (mouth) of *Hydra* polyps (*Bode, 2011*; *Meinhardt, 1993*; *Vogg et al., 2019*), and provides an inhibitory signal that spreads through the umbrella and prevents the induction of additional manubria. We tested this hypothesis by grafting supernumerary manubria at different subumbrellar positions. Grafted manubria could co-exist stably with the original one (*Figure 7A*), with each manubrium behaving independently and participating in feeding. The grafted manubria rapidly sprouted new canals, which reconnected to the existing gastrovascular system (*Figure 7A*). Grafting a fragment of a manubrium base onto the umbrella tissues, either on a radial canal or directly adjacent to the endogenous manubrium, systematically led to its regeneration into a functional extra manubrium (*Figure 7B*; tubular/bi-lobed shape: 12/12). Furthermore, manubria grafted either on the radial canal (*Figure 7C*) or the umbrella (*Figure 7D*) did not prevent the regeneration of an excised manubrium, resulting in the formation of a two-manubria medusa (n: 10/10 and 17/20, respectively). In contrast, no regeneration occurred when exogenous manubria were grafted adjacent to the excised one (*Figure 7E*; n: 15/20). These experiments indicate that existing manubria can exert only a local inhibition on manubrium reformation.

The organization of muscle fibers in manubrium-grafted jellyfish provided a possible explanation for the observed short-range inhibitory influence (*Figure 7F–H*). Grafted manubria appeared to provoke a local reorganization of smooth muscle fibers, which converged towards the grafted manubrium (*Figure 7F–H*). Significantly, in cases where the grafted manubria were positioned close to the ablation site of the endogenous manubrium (resulting in an inhibition of its regeneration), the smooth muscle fibers were not arranged in a hub at the ablation site, but laid parallel to the wound site. These grafting experiments reinforce the systematic correlation between the presence of a radial muscle hub and the regeneration of a new manubrium. They also suggest that local disturbance of muscle fibers orientation by a grafted manubrium plays a role in inhibiting manubrium regeneration (see also *Figure 9—figure supplement 1*).

These experiments indicate that the manubrium does not generate a long-range signal affecting organ positioning. The 'inhibitory' effect of proximal grafts is likely indirect, and can be explained by a localized rearrangement of the muscle fibers. Also on a local scale, we observed new canals sprouting from the base of grafted manubria (*Figure 7A*). The manubrium of *Clytia* medusae, unlike the hypostome of *Hydra* polyps, thus does not appear to act as an organizer of global patterning.

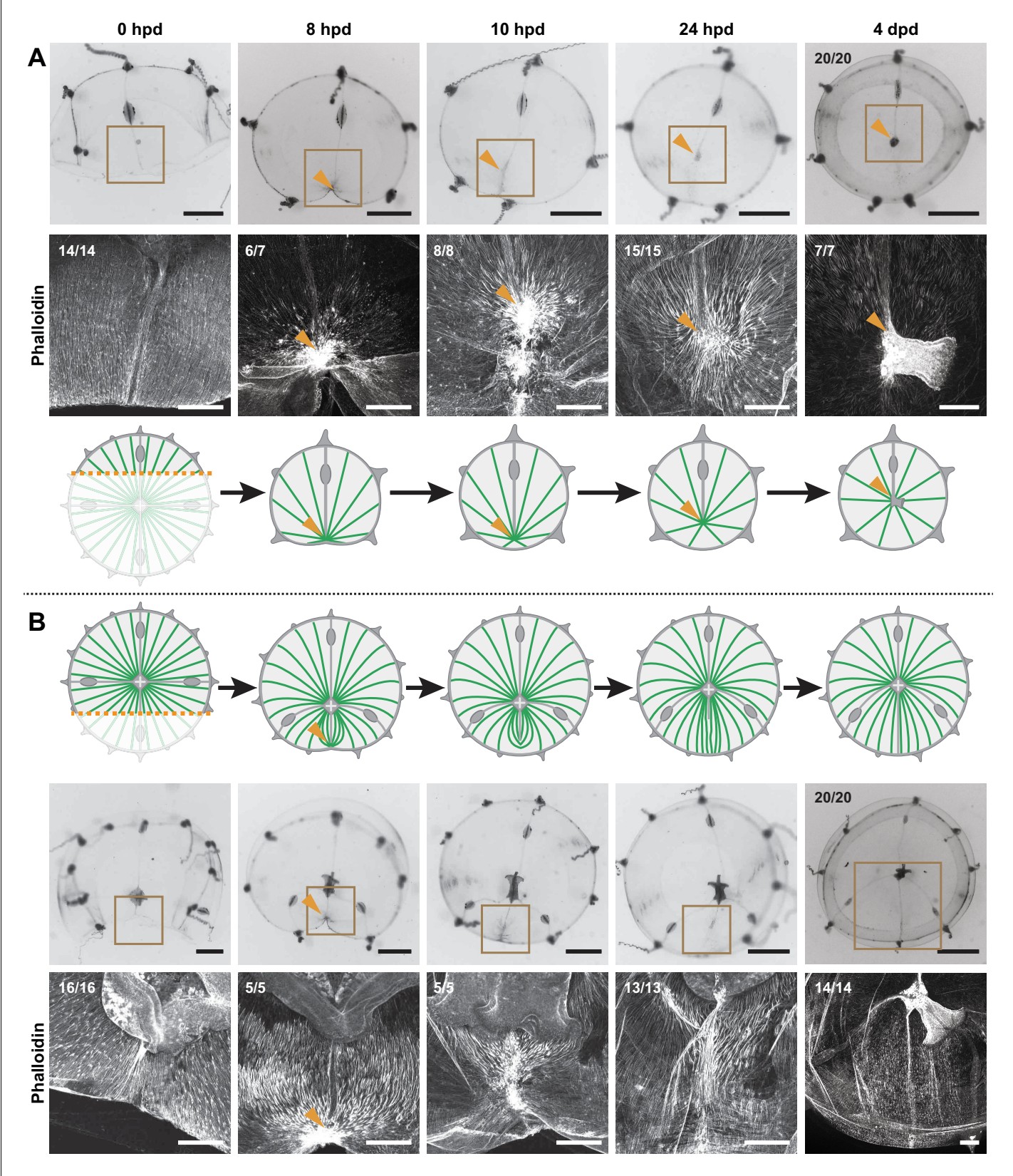

**Figure 9.** Hub stabilization is required for manubrium blastema formation. (**A,B**) Successive images (0, 8, 10, 24 hpd and 4 dpd) of remodeling jellyfish fragments, deriving from an asymmetric cut across the umbrella, which generates one smaller fragment ('S') without manubrium (**A**) and one larger

*Figure 9 continued on next page*

*Figure 9 continued*

fragment ('L') bearing a manubrium (**B**). For each case it is shown: stereomicroscope images of immobilized specimens, phalloidin staining (in white) of the region highlighted in the brown square, diagram showing the topology of radial smooth muscle fibers (in green). All S and L fragments displayed, respectively, the same morphological progression (n: 20/20 each condition). Orange arrowheads indicate hubs. Scale bar: 200 μm, except for stereomicroscope images: 1 mm.

The online version of this article includes the following figure supplement(s) for figure 9:

**Figure supplement 1.** Muscle hub dynamics in proximal manubrium grafts.

Rather it can be considered the pivotal element of a self-organizing system based on local interactions.

## Actomyosin-driven subumbrella remodeling restores medusa shape

Wound closure progressively constricts injured tissues, drawing together the intact parts (*Figure 8A–C*). In all cases, a continuous bundle of actin lined the remodeling edge (*Figure 8F*), reminiscent of the actin ring described around wound sites in the exumbrella layer of *Clytia* (*Kamran et al., 2017*). This bundle resembles the purse-string structures described in many species, tightening epithelial wounds through a contraction of supra-cellular actin/myosin cables assembling along the wound edges (*Begnaud et al., 2016*; *Schwayer et al., 2016*). Detection of phosphorylated (activated) Myosin Regulatory Light Chain 2 (MRLC) around the wound area by antibody staining (*Figure 8G* and *Figure 8—figure supplement 1A*) is consistent with local activation of the actomyosin contraction system. Actomyosin cables could be detected following any type of wound in *Clytia* (*Figure 8F*). Treatment with myosin II inhibitors (such as blebbistatin or BDM) impaired umbrella remodeling, providing further evidence that the forces that underpin remodeling of the umbrella result from actomyosin activity (*Figure 8H* and *Figure 8—figure supplement 1B*).

Staining of F-actin and tracing using the vital dye DiI demonstrated that remodeling is accompanied by localized reorganization of both the subumbrellar radial muscle layer (*Figure 8B,C*) and of the underlying mesoglea (*Figure 8D,E*), reorganization that is limited to the area proximal to the wound. During the constriction phase (1–12 hpd), the radial alignment of smooth muscle fibers loosened and they partially disorganized (*Figure 8B*; 6 hpd), recovering a radial orientation later (12–24 hpd) (*Figure 8B,C*). Similarly, the redistribution of DiI droplets indicated that regions distal to the wound were unaffected, while the proximal area accompanied the constricting tissues. In bisected jellyfish DiI staining on the cut margin is found around the newly centered manubrium (*Figure 8D,E*).

A comparable process described in damaged *Aurelia* ephyrae was proposed to depend on rhythmic contractions of the striated muscles (*Abrams et al., 2015*), and indeed newly fragmented *Clytia* undergo vigorous contractions (*Figure 8—figure supplement 1C*). Treatment with the anesthetic menthol, which efficiently blocks umbrella contractions (*Figure 8—figure supplement 1C*), did not prevent remodeling of bisected medusae, inducing only a slight delay of remodeling (*Figure 8I*). Thus rhythmic contractions of the striated muscle are not necessary to the remodeling process in *Clytia*.

## The topology of muscle fibers predicts the formation of a pro-blastema stable hub

Blastema formation always correlates with the stereotypical bunching up of smooth muscle fibers into a hub-like configuration caused by wound healing/remodeling – however not every initial hub leads to a blastema. Which parameter(s) will determine whether a wound is going to induce the regeneration of a new manubrium? The asymmetric bisection of the jellyfish umbrella, which leaves a smaller (S) fragment devoid of manubrium and a larger (L), manubrium-bearing, one, provides an informative case study in which the S-fragment regenerates a manubrium, while the L-fragment does not (*Figure 9*). These opposite outcomes cannot be attributed to the length of the wound nor, as previously shown, to a long-range inhibitory signal from the existing manubrium (see *Figure 7*). Following bisection, both fragments displayed the typical bunching of injured smooth muscle fibers near the wound area (8 hpd, F-actin staining; *Figure 9*). In the S-fragment, smooth muscle fibers subsequently started to converge towards the 'hub' site, maintaining their connections to the margin of the umbrella. This characteristic topology evolved into a stable radial hub structure, which was then

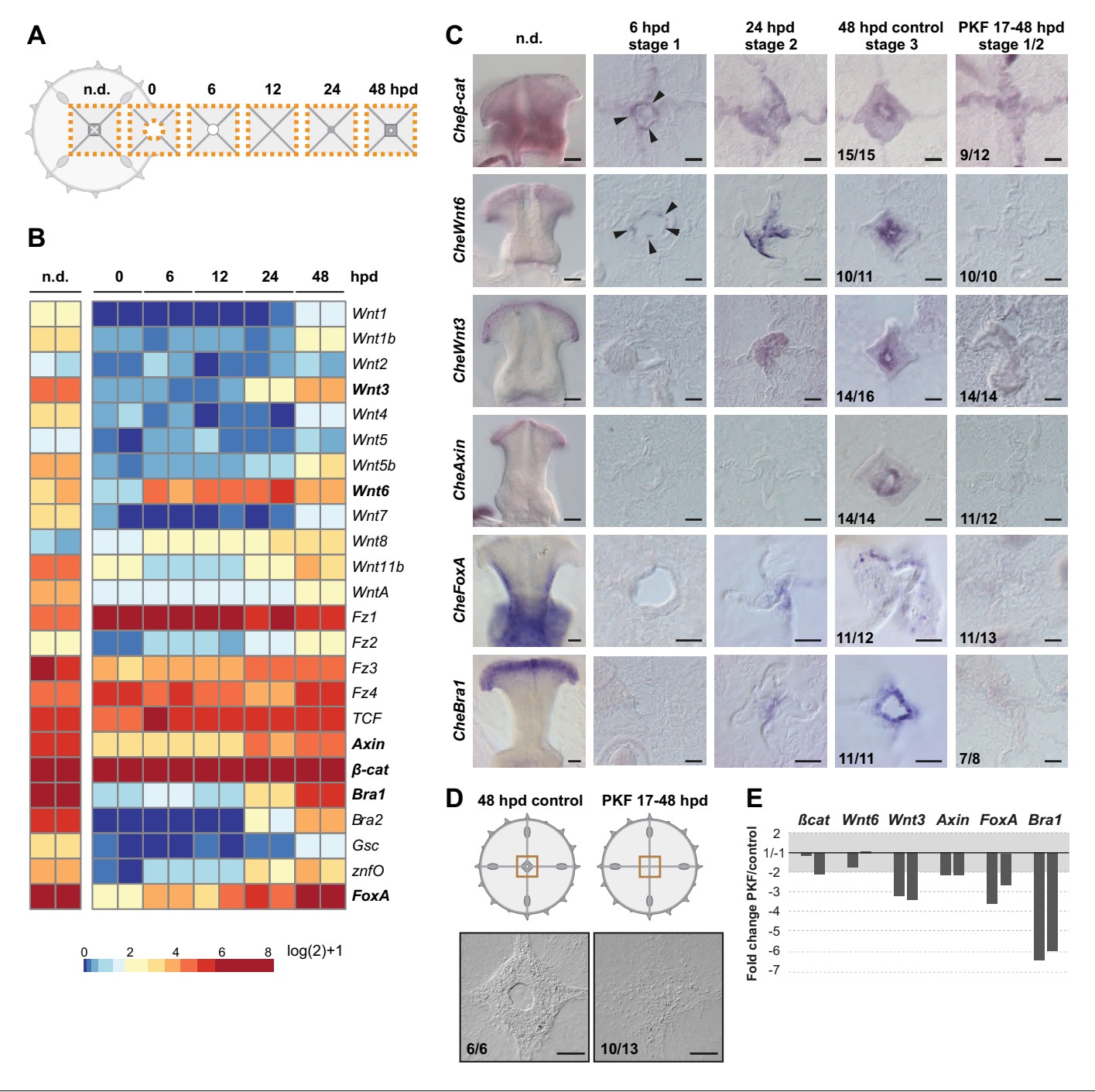

**Figure 10.** Wnt/ß-catenin signaling drives manubrium regeneration. (**A**) Sampling strategy for transcriptomic analyses: fragments of the central umbrella (orange dotted area) were collected at different times, prior (n.d.) or following manubrium ablation. (**B**) Heatmap displaying quantification of transcript levels (tpm, log2 normalized) for selected Wnt/ß-catenin pathway components and downstream targets. Color scale based on quantile values. (**C**) In situ hybridization detection of Wnt pathway components, together with *CheFoxA* and *CheBra1* (genes in bold in B). Genes are expressed in staggered domains along the oral-aboral axis of the undamaged manubrium. *Cheß-cat* and *CheWnt6* transcripts can be detected at the wound site at 6 hpd (black arrowheads), while *CheWnt3*, *CheFoxA* and *CheBra1* are detected only at the stabilized hub (24 hpd). Treatment with PKF118-310 (0.8 μM) inhibits expression of Wnt/ß-catenin pathway components, and their targets. (**D**) Schematics and DIC images of manubrium-ablated jellyfish, treated with PKF118-310 (right) or control (left): following drug treatment blastema did not form and manubrium did not regenerate. (**E**) Reduction (expressed as fold-change) of expression of some Wnt/ß-catenin pathway components, and their targets, determined by qPCR between PKF118-310 (17–48 hpd)

*Figure 10 continued on next page*

*Figure 10 continued*

and control non-treated medusae (48 hpd) (see Materials and methods for sampling details), for two biological replicates. The ratio of expression levels of selected genes were normalized with respect to *eF1α*.

The online version of this article includes the following source data and figure supplement(s) for figure 10:

**Source data 1.** Raw expression data used to construct the heatmap shown in *Figure 10B*; information about the RNAseq samples; expression values (tpm) for all the *Clytia hemisphaerica* transcripts for the 12 RNAseq samples.

**Source data 2.** Primers and raw data of the qPCR experiments shown in *Figures 10E* and *11C* and *Figure 11—figure supplement 1*.

**Figure supplement 1.** PCA plot of RNAseq samples from the manubrium area.

**Figure supplement 2.** PKF118-310 treatment of embryos.

translated towards the new geometrical center of the umbrella, prefiguring the appearance of a manubrium primordium (*Figure 9A*). In the L-fragment, the severed fibers that gathered into the initial hub had a different organization and instead adopted a parallel arrangement, all being connected at one end to the wound site and at the other extremity to the manubrium (*Figure 9B*). During subsequent re-centering of the manubrium, this hub figure displaced away from the wound site, and the surrounding fibers reoriented to reestablish the original configuration around the endogenous manubrium, while the wound-induced hub disappeared (*Figure 9B*).

A similarly transient hub was observed in specimens where manubrium regeneration was inhibited by a locally grafted manubrium as described above (*Figure 9—figure supplement 1*, see also *Figure 7H*). In these experiments, the smooth muscles initially aggregated into a hub at the wound site (12 hpd), but later disassembled, accounting for the absence of regenerative blastema at 48 hpd (and related hub).

We propose a model where the stabilization of a wound-induced hub depends on the configuration of the connecting muscle fibers: stabilization of the muscle hub occurs when the fibers are not constrained by attachment to an existing hub.

## Wnt signaling links muscle hub stabilization to manubrium regeneration

We used an RNAseq approach to provide molecular insights into the regulation of *Clytia* regeneration, following the transcriptional dynamics of the central umbrella region at different times during manubrium regeneration (*Figure 10A* and *Figure 10—figure supplement 1*). Transcript levels of Wnt-pathway components and of known downstream targets (*CheBra1* and *CheFoxA*, see *Lapébie et al., 2014*) modulated dynamically during the regeneration process (*Figure 10B*). These are prime candidate regulators of manubrium regeneration given their widespread role in developmental regenerative/processes, and notably in determining oral fates in cnidarians (e.g. *Duffy et al., 2010*; *Momose et al., 2008*; *Servetnick et al., 2017*). Most of *Clytia* Wnt genes were found to be expressed highly in the non-dissected manubrium (n.d.), and lowly in the surrounding umbrella tissues (0 hpd) (*Figure 10B,C*). Their expression then increased during manubrium regeneration (*Figure 10B,C*). In situ hybridization revealed that, following manubrium excision, *CheWnt6* and *Cheβ-cat* were activated early at the constricting wound area (6 hpd), while transcripts for *CheWnt3* and the Wnt-pathway downstream targets *CheBra1* and *CheFoxA* started to be detectable later at the 'hub' site (24–48 hpd; *Figure 10B,C*). Expression of *CheAxin*, a modulator of canonical Wnt signaling, could first be detected at the hub/blastema site from 48 hpd (*Figure 10C*).

Treatment of injured jellyfish (from 17 hpd until 48 hpd) with PKF118-310, a chemical inhibitor of Wnt/β-catenin signaling, blocked regeneration of manubrium prior to blastema development ('stages 1–2' in *Figure 10D*). Inhibition of Wnt/β-catenin signaling by PKF118-310 was validated using *Clytia* embryos, where the role of this pathway in axis development is well characterized (*Momose et al., 2008*; *Lapébie et al., 2014*). 21 hr treatments, starting from cleavage stage, phenocopied the strong 'aboralization' obtained following *CheWnt3* knockdown (*Momose et al., 2008*). No gastrulation or signs of morphological polarity were observed 24 hr after fertilization (24 hpf; *Figure 10—figure supplement 2*), while embryos washed from the drug recovered and formed polarized larvae (*Figure 10—figure supplement 2*).

In dissected jellyfish, treatment with PKF118-310 prevented expression of Wnt-pathway components and downstream targets (*CheWnt3, CheWnt6, CheBra1, CheFoxA, CheAxin)* at the hub, as shown by in situ hybridization (*Figure 10C*). qPCR experiments showed a strong reduction of the

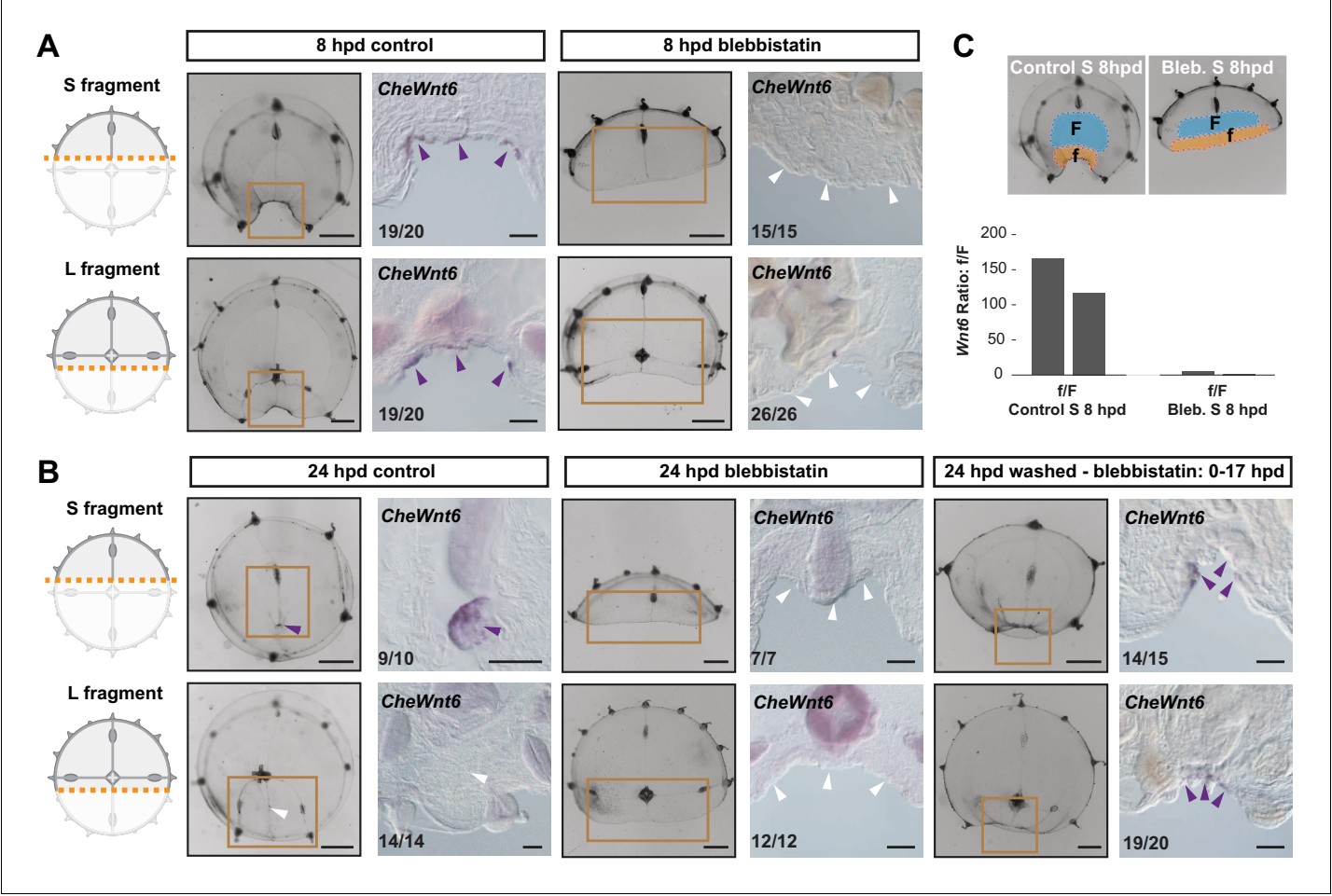

**Figure 11.** Remodeling-dependent activation of *CheWnt6* expression. (**A**) Schematics of S/L fragments to the left, and stereomicroscope/in situ hybridization images of 8 hpd control and blebbistatin-treated to the right. At 8 hpd *CheWnt6* expression was detected at the remodeling wound site of both fragments (purple arrowheads). In blebbistatin-treated samples, no *CheWnt6* expression was detected at the wound sites (white arrowheads). (**B**) At 24 hpd *CheWnt6* was not detected in the L fragments at the site of hub disassembly (white arrowheads), while it was detected at the hub site in S fragments (purple arrowhead). No *CheWnt6* expression was detected at the wound sites in blebbistatin-treated samples at 24 hpd, while *CheWnt6* signal was detected following washout of blebbistatin (blebbistatin treatment: 0–15 hpd) at the remodeling wound site (at 24 hpd), in both S and L fragments. (**C**) *CheWnt6* expression ratio between 'f' (corresponding to wound area) and 'F' (nearby umbrella area) tissue pieces, as determined by qPCR, for control and blebbistatin-treated S fragments (8 hpd), for two biological replicates normalized with respect to *eF1α* expression. See Materials and methods and *Figure 11—figure supplement 1* for further details, and *Figure 10—source data 2* for the raw qPCR data. (**A,B**) Scale bars: 100 µm, except for stereomicroscope images: 1 mm.

The online version of this article includes the following figure supplement(s) for figure 11:

**Figure supplement 1.** Treatment with blebbistatin inhibits early increase of *CheWnt6* expression at wound area.

target genes (*CheBra1*, *CheFoxA*), a milder reduction for some pathway components (*CheWnt3*, *CheAxin)*, and no significant reduction for the early-expressed genes *CheWnt6* and *Cheβ-cat* (*Figure 10E*). When PKF118-310 treatment commenced immediately after dissection, it strongly delayed wound closure, suggesting that Wnt/β-catenin signaling may also be involved in early phases of remodeling.

## Remodeling-dependent *Wnt6* expression

In situ hybridization on 'S' and 'L' fragments from bisected medusae revealed an early activation of *CheWnt6* expression on both 'sides' of the remodeling wound (8 hpd; *Figure 11A*). Following the subsequent disassembly of the hub in the larger fragments, *CheWnt6* expression became undetectable by in situ hybridization in L fragments, while it reinforced in the S fragments, where hubs were

stabilized (24 hpd; *Figure 11B*). Inhibition of remodeling with blebbistatin suppressed *CheWnt6*-expression in both S and L fragments (at 8 hpd and 24 hpd; *Figure 11A,C*), while *CheWnt6* signal was detected at the remodeling wound site (at 24 hpd) following 9 hr of inhibitor washout, in both S and L fragments (*Figure 11B*). qPCR quantification revealed a much stronger upregulation (>100-fold) of *CheWnt6* in the remodeling wound area of untreated S fragments at 8 hpd (*Figure 11C* and *Figure 11—figure supplement 1*) compared to the relaxed wound area of blebbistatin-treated S fragments (2- to 4-fold) (*Figure 11C* and *Figure 11—figure supplement 1*). Blebbistatin treatment also caused an unexpected generalized increase in *CheWnt6* expression in the umbrella tissues of both non-dissected and injured jellyfish (*Figure 11—figure supplement 1*). These analyses point to a functional link between actomyosin-driven remodeling and the expression of Wnt-pathway components.

Taken together, our data suggest that expression of *Wnt6* is strongly and rapidly activated during actomyosin-dependent wound remodeling in *Clytia,* and later maintained at the stabilized muscle hub. Wnt/β-catenin signaling becomes necessary for later stages of manubrium regeneration.

## Discussion

*Clytia* jellyfish, with their tetraradial body elaborated around a short oral-aboral axis, and defined organs harboring stem cell pools, provide a novel paradigm for the regulation of regenerative processes and body patterning in adult forms. *Clytia* jellyfish can cope efficiently with a wide range of perturbations, rapidly regaining functionality and a stable body organization. Our study shows that their regenerative potential relies on the coordinated interplay of tissue re-organization, proliferation of cellular progenitors and long-range cell recruitment (summarized in *Figure 12*). The global pattern of regenerating *Clytia* emerges from the local interactions of structural elements, and mechanically-driven tissue remodeling controls the regenerative process: in fragments lacking a manubrium, reorganization of smooth muscle fibers into a 'hub' provides a structural landmark for Wnt signaling-dependent blastema formation. Cues from the gastrovascular canals further sculpt manubrium morphogenesis.

### Distinct cell behaviors, coordinated in time and space

Regeneration in *Clytia* jellyfish is characterized by large-scale remodeling (see *Figure 12*): an initial re-organization of umbrella tissues, triggered by wound healing, leads to body recircularization (*phase i*), and continues with global re-positioning of the organs and canal system (*phase ii*). The timing and site of blastema formation is determined precisely. The manubrium blastema, a heterogeneous cell mass, recruits distinct cell types from nearby organs. These include both putative stem cells expressing *CheNanos1*, and differentiated cells – notably a novel type of gastrodermal cell that we term Mobilizing Gastro-Digestive (MGD) cells. MGD cells remain largely uncharacterized, but their mobilization between organs may contribute to the redistribution of nutrients. The stem cell system(s) of adult *Clytia* jellyfish have not been fully characterized. Only a fraction of proliferating cells express the i-cell marker *CheNanos1*, suggesting that in *Clytia* several distinct stem cell populations may co-exist, as in *Hydra* (e.g. see *Bosch, 2009*).

The relative contributions of cell proliferation and tissue remodeling to the restoration of form vary widely among regenerating species. In the classic regeneration model *Hydra*, as in *Clytia*, body regeneration mainly involves remodeling and repatterning of existing tissues (*Bode, 2003*; *Vogg et al., 2019*). In contrast, cell proliferation plays a prominent role during regeneration of *Nematostella* and *Hydractinia* polyps (*Amiel et al., 2015*; *Bradshaw et al., 2015*; *Passaneck and Martindale, 2012*). The few studied cases of organ regeneration in cnidarians also require cell proliferation, as here shown in *Clytia* or in the regenerating tentacles of some hydrozoan jellyfish (*Fujita et al., 2019*). Targeted recruitment of undifferentiated progenitors towards the wound/regeneration site is also a widespread feature of regenerating systems, including *Hydractinia* polyps (*Bradshaw et al., 2015*), planarians (*Atabay et al., 2018*), or mammalian epithelia (e.g. *Dekoninck and Blanpain, 2019*).

### Mechanical control of shape in regenerating *Clytia*

Historical experiments on specimens caught from the wild revealed several basic features of *Clytia* regeneration, notably the rapid re-circularization of jellyfish fragments and the correlation between

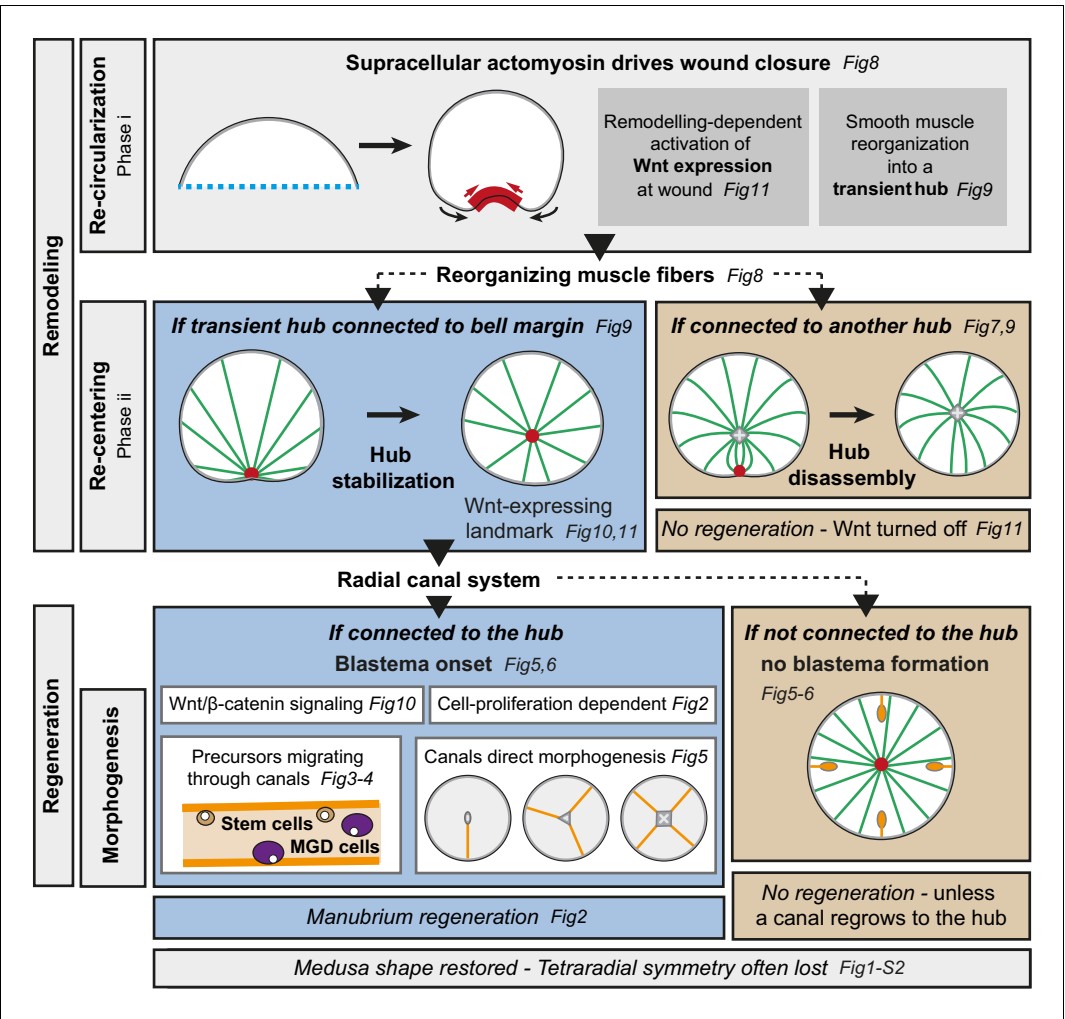

**Figure 12.** Summary figure. Actomyosin-driven remodeling restores the circular shape (*phase i*), and generates a transient hub of remodeling smooth muscle fibers expressing *CheWnt6*. If stabilized after the re-centering step (*phase ii*), the hub maintains *CheWnt6* expression and transforms into a Wnt-signaling landmark, enabling blastema initiation. Manubrium regeneration is fueled by cells traveling through at least one radial canal. The circular medusa shape is almost systematically restored, but the original tetraradial symmetry is often lost after completion of remodeling and regeneration.

the shape of the fragment and both its regeneration potential and its ability to recover the original tetraradial symmetry (*Neppi, 1918*; *Schmid and Tardent, 1971*; *Schmid, 1974*; *Schmid et al., 1976*). In hydrozoan polyps, the mouth (hypostome) acts as a signaling center and the source of a morphogen gradient organizing the body axis (e.g. *Bode, 2009*; *Mayorova et al., 2015*). In contrast, morphogen gradients are not responsible for the positioning and regeneration of manubria in *Clytia* jellyfish, as shown by our grafting experiments and the pioneering work of *Schmid et al., 1976*. Similarly to our findings, they observed an inhibitory effect when a manubrium (or fragment of it) was grafted close to the regeneration site, but they explained it as a competition for the cells contributing to regeneration (*Schmid et al., 1976*). Their model for regeneration relied on mechanical forces rather than on a morphogen gradient, and more precisely postulated that the tensions exerted by the mesoglea and the smooth/striated muscle fibers would either concentrate, or favor the dispersion of the subumbrellar cells contributing to manubrium regeneration (*Schmid et al., 1976*). Indeed, we found that the regeneration of a manubrium is related to the organization of the smooth muscle fibers: regeneration is promoted when the muscle topology - and thus presumably the balance of resultant forces - allows the stabilization of a 'hub' conformation. As for the cellular components, we found a body-wide contribution from the jellyfish, with the highly plastic canal

system providing a conduit for the diverse cell types fueling manubrium regeneration. The mesoglea, the elastic layer constituting a large part of the jellyfish body, undoubtedly contributes to the mechanical properties of the system, likely by opposing the muscle forces. Consistently, in vitro experiments using another jellyfish species, *Podocoryna*, showed that mechanical deformation of the mesoglea affects the spreading of striated muscle cells and the initiation of DNA replication (*Schmid et al., 1993*).

We found that the rapid re-circularization of jellyfish fragments relies on the assembly of an actomyosin cable at the wounded edge, leading to the contraction of the cut segments and re-establishing tissue integrity. Supra-cellular cables generated by the alignment of cortical actomyosin are a widespread feature of unsteady animal tissues, and provide a conserved mechanism for the closure of epithelial gaps (*Jain et al., 2019*). Actomyosin cables have been broadly reported during wound healing and morphogenesis (reviewed in *Röper, 2013*; *Schwayer et al., 2016*), including in the simple epithelial tissue that covers the *Clytia* jellyfish exumbrella (*Kamran et al., 2017*).

During regeneration of *Clytia* fragments, pharmacological inhibition of actomyosin contractility prevented not only tissue remodeling, but also *CheWnt6* expression at wound and blastema formation. Our findings thus suggest that Wnt/β-catenin signaling activation is modulated by actomyosin contractions. Such instances of biochemical responses triggered by mechanical forces are attracting increasing attention (*Chiou and Collins, 2018*; *Green and Sharpe, 2015*; *Ingber, 2006*; *Urdy, 2012*; *Vining and Mooney, 2017*). Interplay between mechanical stresses and gene expression allows a fine coordination of developmental processes (e.g. *Heisenberg and Bellaïche, 2013*), with numerous examples identified during embryogenesis such as mechanical regulation of Wnt/β-catenin signaling during germ layer formation in *Drosophila and* zebrafish (*Brunet et al., 2013*), amphibians (*Kornikova et al., 2010*), *Nematostella* (*Pukhlyakova et al., 2018*), or during antero-posterior body axis establishment in mammals (*Hiramatsu et al., 2013*). Supra-cellular actomyosin fibers may also be involved in body axis repositioning during regeneration in *Hydra* polyps (*Livshits et al., 2017*; *Mercker et al., 2015*), although it remains unclear how mechanical cues are integrated with the morphogen gradient that patterns the primary body axis during both homeostasis and regeneration (*Wang et al., 2020*; *Nakamura et al., 2011*; *Hobmayer et al., 2000*; *Gierer and Meinhardt, 1972*).

## The emergence of a Wnt-expressing structural landmark for blastema formation

Wound sites and the neighboring tissues act as signaling centers in many regenerating organisms, instructing the re-patterning of body parts or blastema initiation – for example in *Hydra* polyps (*Chera et al., 2009*), zebrafish caudal fins (*Wang et al., 2019*), or planarians (*Oderberg et al., 2017*; *Petersen and Reddien, 2009*; *Reddien and Sánchez Alvarado, 2004*; *Wurtzel et al., 2015*). We found an analogous correlation between wound site and blastema position, however in *Clytia* medusa signaling from the wound site is not sufficient to trigger regeneration, but it is likely conditional. This conclusion is supported on one hand by the fact that wound response and blastema onset can be temporally separated, as also recently shown for the regeneration of the ctenophore *Mnemiopsis* (*Ramon-Mateu et al., 2019*), and on the other that the trigger for regeneration depends on the configuration of the wound.

In *Clytia* jellyfish, wound closure gathers the damaged smooth muscle fibers into a transient structure, termed the 'hub'. The convergence of multiple 'spokes' (fibers) around this hub precisely positions the regeneration site. If stabilized, the hub becomes a *CheWnt6*-expressing landmark, enabling the development of the manubrium blastema. The hub is therefore not merely an anatomical structure, but actively maintains Wnt signaling. Conversely, if smooth muscle fibers converge on a pre-existing hub (e.g. towards a manubrium), the wound-induced one will disassemble, *CheWnt6* expression will cease at this location and no manubrium regeneration will occur. Manubrium regeneration requires sustained Wnt-pathway signaling, as reported for several other regenerating systems – such as axolotl limbs (e.g. *Yokoyama et al., 2007*) or planarian heads (*Petersen and Reddien, 2009*).

The mechanisms regulating the stabilization and final positioning of the 'spokes' and the 'hub' remain to be fully established. The local alignment of myofibrils might follow cell polarity cues, likely under the control of PCP (*Seifert and Mlodzik, 2007*), previously shown to orient epithelial cells during *Clytia* larval development (*Momose et al., 2012*). While the initial gathering of spokes is a consequence of the wound-healing process, the reorganization of smooth muscles into a 'spokes and hub'

configuration is likely also guided by mechanical tensions generated by the intact muscles anchored to the umbrella margin. Post-wound topology of muscle fibers is a reliable predictor of the 'hub' fate, strongly suggesting a role for mechanical stresses in its stabilization.

'Spoke and hub' muscle configurations equivalent to the one we describe here in *Clytia* can also be recognized during wound healing/regeneration in *Hydra* polyps (*Livshits et al., 2017*) and planarians (*Scimone et al., 2017*), even though they have not been pointed out explicitly. In planarians, the position of the anterior pole in intact and regenerating animal correlates with the meeting point of the longitudinal muscle fibers, preceding the appearance of a new mouth or blastema. 'Spoke and hub' patterning systems might thus represent a general but overlooked feature of regenerating animals, playing an early role in the stabilization of remodeling tissues and acting as 'organizers', accurately positioning structures in broadly perturbed systems.

During regeneration in *Clytia*, the configuration of the muscle hub and its associated spokes represents a good readout for the regenerate pattern, as it reliably predicts the position of the pro-blastema Wnt/β-catenin signaling center. The mechanisms linking muscles to the newly developing tissue polarity of the regenerate remain however unclear. The assembling spokes could carry/generate polarity information which is transmitted to the tissues, or their positioning may be directed by another patterning cue in the tissue. The question whether muscle fibers contribute to the polarity of a tissue – by exercising contractile forces – or they align according to another morphogenetic signal – for example responding to Wnt-pathway signaling – is not restricted to *Clytia*, but is relevant also for other systems, such as *Hydra* (*Wang et al., 2020*). Since whole-body morphogen gradients do not appear to contribute to manubrium positioning in the medusa of *Clytia*, this study system could be advantageous to disentangle the relative contributions of muscle fibers and biochemical signaling to body patterning.

## Emergence of global pattern from local interactions

How positional information is restored in animals undergoing regeneration remains a central question. In *Clytia*, morphological 'quadrants' (each containing a radial canal, a quarter manubrium, a gonad and a segment of the bell rim with tentacle bulbs) repeat around the primary oral-aboral body axis. Our findings demonstrate that, in regenerating animals, the angular position of elements around this body axis is not encoded by a system of rotational coordinates, but depends rather on cues from existing structures. The characteristic tetraradial symmetry of the adult jellyfish thus does not depend on an actively maintained global patterning system. We cannot nevertheless exclude that a system of morphogen might be secondarily patterning the centro-peripheral axis: missing gonads, for instance, reform at their original location along the radial canal. Manubrium regeneration, instead, is essentially directed by the reorganization of radial muscles and the local influences of the canal system. The integration at the jellyfish level of local cues, emerging from the interaction of structural elements, is thus sufficient to explain body shape recovery and manubrium regeneration - without invoking a redefinition of positional information, encoded by a body-wide, morphogen-based, patterning system.

## Materials and methods

**Key resources table**

| Reagent type (species) or resource | Designation | Source or reference | Identifiers | Additional information |
|---|---|---|---|---|
| Gene (*Clytia hemisphaerica*) | *CheNanos1* | GenBank | JQ397274 | |
| Gene (*Clytia hemisphaerica*) | *CheWnt3* | GenBank | EU374721 | |
| Gene (*Clytia hemisphaerica*) | *CheWnt6* | GenBank | EU374719 | Named *CheWnt6* following *Condamine et al., 2019* |
| Gene (*Clytia hemisphaerica*) | *CheAxin* | GenBank | EU374716 | |

*Continued on next page*

*Continued*

| Reagent type (species) or resource | Designation | Source or reference | Identifiers | Additional information |
|---|---|---|---|---|
| Gene (*Clytia hemisphaerica*) | *Cheβ-cat* | GenBank | JQ438997 | |
| Gene (*Clytia hemisphaerica*) | *CheBra1* | GenBank | DQ872898 | |
| Gene (*Clytia hemisphaerica*) | *CheFoxA* | GenBank | GBGP01000196 | |
| Gene (*Clytia hemisphaerica*) | Z4B strain | *Leclère et al., 2019* | | Polyp colony producing female jellyfish |
| Antibody | Rhodamine goat anti-rabbit secondary antibody | Jackson Immuno Research | | IF (1:500) |
| Antibody | Alexa Fluor 488 or 594 goat anti-rat secondary antibodies | Thermo Fisher Scientific | #A-11006 #A-11007 | IF (1:200) |
| Antibody | anti-Myosin Light Chain two phosphorylated at serine19 (PMyo) rabbit polyclonal | Cell signaling technology | #3671 | IF (1:200) |
| Antibody | anti-tyrosinated tubulin antibody (YL1/2) rat monoclonal | Abcam | #6160 | IF (1:50) |
| Chemical compound, drug | Hydroxyurea | Sigma-Aldrich | #H8627 | |
| Chemical compound, drug | Blebbistatin | Sigma-Aldrich | #B0560 | |
| Chemical compound, drug | BDM (2,3-Butanedione monoxime) | Sigma-Aldrich | #B0753 | |
| Chemical compound, drug | Menthol | Sigma-Aldrich | #M2772 | |
| Chemical compound, drug | PKF118-310 | Sigma-Aldrich | #K4394 | |
| Commercial assay, kit | EdU Click-It kit (Alexa Fluor 555) | Thermo Fisher Scientific | #C10338 | |
| Other | FluoSpheres | Molecular Probes | #F8812 | |
| Other | DiI | Thermo Fisher Scientific | #D3911 | |
| Other | Alexa Fluor 488 Phalloidin | Thermo Fisher Scientific | #10125092 | (1:100) |
| Other | Hoechst dye | Sigma-Aldrich | #94403 | (1 µg/ml) |
| Other | Reference transcriptome (*Clytia hemisphaerica*) | *Leclère et al., 2019* | ENA identifier: PRJEB28006 | Genome guided transcriptome |
| Other | RNAseq samples (*Clytia hemisphaerica*) | This study | ENA identifier: PRJEB37920 | BGISEQ-500 50nt single reads generated for 12 RNA samples: pooled samples of umbrella central area collected at six time points during manubrium regeneration (two biological replicates for each). |

## *Clytia hemisphaerica* husbandry

*Clytia* jellyfish were produced clonally from the Z4B (female) polyp colony (see *Leclère et al., 2019* for further information on the genome and the establishment of the line) in the *Clytia* facility at IMEV, as described by *Lechable et al., 2019* using a custom-made closed culture system, artificial sea water (37 ‰, 'Red Sea Salts' (Red Sea) in deionized water) at 18°C, and twice-daily feeding with *Artemia sp.* nauplii. We performed all experiments on 11/14 day old, newly spawning medusae (0.5–0.8 cm diameter). Animals were starved for at least 12 hr prior to experiments and not fed during

regeneration experiments, with the exception of the specified gonad and bulb-regeneration experiments and of the fluorescent bead assays (see below).

## Surgical procedures

Surgical manipulations were performed in agarose-coated petri dishes (2% agarose in Millipore-Filtered Artificial Sea Water, hereby FASW), by means of custom-made tools of fine platinum wire, acupuncture needles, pipette tips of diverse diameters, Dowell scissors (Fine Science Tools, #15040–11), fine forceps (Fine Science Tools, #11370–40) and fine scalpels. For manipulations, animals were relaxed using either ice-cold FASW or 400 µM menthol in FASW. Tentacles were trimmed with fine scissors to avoid damage to the animals due to entanglement. Operated animals were maintained in clean 6-well plastic plates, in a volume of 5 ml of FASW with antibiotic (1/1000 dilution of stock, stock at 10,000 units penicillin and 10 mg streptomycin per ml; Sigma Aldrich, #P4333), refreshed at least once per day. Multi-well plates were kept in an incubator at 18°C.

Organ dissections were performed either by punching out a hole across subumbrella, mesoglea and exumbrella using an appropriately-sized pipette tip, or by excising the targeted element using forceps and a scalpel or with custom-made wire tools. Grafts were performed using acupuncture needles to stitch fragments together, and fine forceps.

Experiments were monitored at least once per day, and pictures were taken when necessary (see below). For imaging, animals were temporarily relaxed using 400 µM menthol in FASW.

## In vivo cell labeling experiments and mesoglea injections

EdU labeling was performed by incubating medusae in a 100 µM EdU (Click-It kit, Thermo Fisher Scientific #C10338) in FASW. Incubation times for pulse and pulse-chase experiments are reported in the Results section and in the related Figures. Animals were then fixed with 4% paraformaldehyde (PFA) in PBS, for two hours, then rinsed with 1X PBS. Staining was performed with the EdU Click-It kit (Alexa Fluor 555 kit; Thermo Fisher Scientific, #C10338) following the manufacturer's protocol (BSA not added). Nuclei were counterstained with 1 µg/ml Hoechst 33258 (Sigma-Aldrich, #94403). Fluorescent bead labeling (FluoSpheres, Molecular Probes, #F8812) was performed by feeding animals with beads mixed with hatched *Artemia* nauplii. Beads were first washed through five successive centrifugations in milli-Q $H_2O$. *Artemia* nauplii were then incubated with the beads for one hour. After feeding, jellyfish were washed with FASW, to remove non-ingested beads. DiI (Thermo Fisher Scientific, #D3911) in Wesson oil was injected into the mesoglea with micro-needles.

## Chemical treatments

Around 10 mM hydroxyurea (Sigma-Aldrich, #H8627) was dissolved in FASW; the solution was renewed twice a day. Muscle contraction was inhibited as follows: with 1 or 5 µM blebbistatin (Sigma-Aldrich, #B0560) diluted from a 34 mM stock solution in DMSO, or with 8 mM BDM (2,3-Butanedione monoxime; Sigma-Aldrich, #B0753) in FASW, or with 400 µM menthol (Sigma-Aldrich, #M2772) diluted from a 1M stock solution in ethanol. The β-catenin/Tcf interaction inhibitor PKF118-310 (*Lepourcelet et al., 2004*; Sigma Aldrich, #K4394) was used at a final concentration of 0.7–0.8 µM, diluted from a 15 mM stock solution in DMSO.

## In situ hybridization, immunostaining and phalloidin staining

Probe synthesis and fluorescent in situ hybridization (FISH) were performed according to our urea-based protocol (*Sinigaglia et al., 2018*), see also *Sinigaglia, 2019* for further recipes and reagents. In situ hybridization probes were generated from clones previously reported: *CheBra1* and *Cheβ-cat* (*Momose and Houliston, 2007*); *CheAxin* (*Momose et al., 2008*); *CheNanos1* (*Leclère et al., 2012*); *CheFoxA* (*Lapébie et al., 2014*); *CheWnt3* and *CheWnt6* (*Condamine et al., 2019*). In situ hybridization experiments were performed at least twice for each condition, and gave highly reproducible results (see respective figures for sample sizes).

For immunostaining, animals were fixed for two hours with 4% PFA in HEPES buffer (HEPES 0.1M, pH 6.9, EGTA 50 mM, $MgSO_4$ 10 mM, Maltose 80 mM), rinsed and permeabilized with 1X PBS containing 0.2% Triton X-100, blocked and incubated with an antibody recognizing Myosin Light Chain two phosphorylated at serine19 (1:200, rabbit; Cell signaling technology, #3671), or with the anti-tyrosinated tubulin antibody YL1/2 (1:50, rat; Abcam, #6160). Primary antibodies were detected,

respectively, with a goat anti-rabbit secondary antibody coupled to rhodamine (1:500) and a goat anti-rat secondary antibody coupled to Alexa 488 or 594 (1:200; Thermo Fisher Scientific, #A-11006 and #A-11007); samples were mounted in Citifluor AF-1 antifade mountant for imaging. Actin fibers were stained with 1:100 phalloidin coupled to Alexa 488 (solubilized in methanol; Thermo Fisher Scientific, #10125092), following fixation with 4% formaldehyde in HEPES buffer. For FISH plus EdU reaction, animals were fixed with 3.7% formaldehyde in HEPES buffer and the EdU click-It reaction was performed after the TSA reaction of FISH. Nuclei were stained with 1 µg/ml Hoechst 33258 (Sigma-Aldrich, #94403).

## Image acquisition and processing

Macroscopic images were taken with a Sony camera (NEX-5R) mounted on an Olympus SZ61 stereomicroscope. Fluorescent images were taken with a Zeiss Axio Imager A2 microscope and a Leica SP8 confocal microscope. Image processing (maximum projection, color display) and quantifications were done with Fiji software. All stereomicroscope images are shown in black and white with color inversion to facilitate jellyfish structure visualization. Drawings and figure construction were done with Illustrator CS6 (Adobe).

## Transcriptomic analyses

RNAseq samples were prepared from 11/12 day old Z4B jellyfish (4–5 mm in size, 12 tentacle bulbs). Animals were first starved for 24 hr. Manubrium dissections were performed using a P200 tip (Sorenson, BioScience). Biological replicates for each sample consisted of pooled tissue fragments corresponding to the umbrella central area (see *Figure 10A*), dissected using ophthalmological scissors (Fine Science Tools, #15000) at the appropriate time: n.d. (non-dissected manubrium), 0, 6, 12, 24 and 48 hpd. Batches of five dissected fragments were collected in lysis buffer (from the RNAqueous micro kit, Thermo Fisher Scientific #AM1931), then quickly vortexed and snap-frezeed in liquid nitrogen (no longer than 15 min from dissection), and stored at −80˚C until further processing.

BGISEQ-500 50nt single reads were generated by BGI (~24 million reads per sample) from mRNA isolated using the RNAqueous-Micro kit (Thermo Fisher Scientific, #AM1931) followed by a DNaseI treatment (RQ1 DNase; Promega, #M6101) for 20 min at 37˚C and purification with the RNeasy MinElute Cleanup kit (Qiagen, #74204). RNA quality was checked with the Agilent 2100 Bioanalyzer. See *Figure 10—source data 1* for further information about the samples. The reads were mapped against a genome-guided reference transcriptome of *Clytia hemisphaerica* (*Leclère et al., 2019* – ENA identifier: PRJEB28006) using the Kallisto R package (*Bray et al., 2016*). Read counts were combined at the gene level using Sleuth R package (*Pimentel et al., 2017* following *Yi et al., 2018*). The RNA-seq data were deposited in ENA (PRJEB37920). Wnt signaling pathway-related genes were retrieved and named following *Leclère et al., 2019* and *Condamine et al., 2019*.

## Quantitative RT-PCR

Total RNA from fragments (central umbrella fragments for the experiments shown in *Figure 10*; 'f' or 'F' fragments for experiments shown in *Figure 11* and *Figure 11—figure supplement 1*) pooled from 20 to 24 jellyfish was extracted using the RNAqueous-Micro kit according to the manufacturer's instructions (Thermo Fisher Scientific, #AM1931). Two biological replicates were generated for each sample type. Genomic DNA was removed by Ambion DNase I treatment (Thermo Fisher Scientific, #AM2222). First-strand cDNA was synthesized from the entire volume of total RNA using the SuperScript VILO cDNA Synthesis Kit (Invitrogen, Thermo Fisher Scientific). Quantitative PCRs were run in quadruplicate, each reaction contained (20 µl final volume): 5 µl cDNA (1/100 or 1/200), 10 µl SYBR Green I Master Mix (Roche Applied Science) and 200 nM of each gene-specific primer. PCR reactions were run in 96-well plates, in a LightCycler 480 Instrument (Roche Applied Science). *eF1α* was used as reference control gene (*Lapébie et al., 2014*). Expression levels were calculated as $2^{-CP}$, with CP (Crossing Point) being the number of cycles required for the fluorescent signal to cross the threshold. Primer sequences and qPCR raw data are provided in *Figure 10—source data 2*.

## Statistics

Histograms were prepared in Excel, while boxplots and heatmaps with R (R 3.6.1) using *ggplot2* (v. 3.2.1; *Wickham, 2016*) and *pheatmap* (v. 1.0.12) packages respectively. Statistical tests were

performed in R (see Figure legends for details). The principal component analysis on RNAseq samples was performed using Sleuth (*Pimentel et al., 2017*).

## Acknowledgements

We thank Josselin Lupette, Sophie Rei-Rosa, Priscilla Freschu, Gwladys Perez and Emma Labis for experimental assistance, and Sophie Collet, Loann Gissat, Alexandre Jan, Axel Duchene, Régis Lasbleiz and Laurent Gilletta for animal maintenance. We thank Philippe Dru for help with the RNAseq analysis. We further thank Yulia Kraus for grafting advice, Séverine Urdy for discussions, and Michalis Averof for comments on an earlier version of the manuscript. This work was supported by two grants from the Agence Nationale de la Recherche to LL, core CNRS and Sorbonne University funding to the LBDV; PhD fellowships for SP from the Ministère de l'Enseignement Supérieur et de la Recherche and the Fondation pour la Recherche Médicale. We thank the Marine Resources Centre (CRBM and PIV imaging platform) of Institut de la Mer de Villefranche (IMEV), supported by EMBRC-France. The French state funds of EMBRC-France are managed by the ANR within the investments of the Future program.

## Additional information

### Funding

| Funder | Grant reference number | Author |
| --- | --- | --- |
| Agence Nationale de la Recherche | ANR-13-PDOC-0016 | Lucas Leclère |
| Agence Nationale de la Recherche | ANR-19-CE13-0003 | Lucas Leclère |
| Fondation pour la Recherche Médicale | FDT201805005536 | Sophie Peron |
| Ministère de l'Enseignement Supérieur et de la Recherche | | Sophie Peron |

The funders had no role in study design, data collection and interpretation, or the decision to submit the work for publication.

### Author contributions

Chiara Sinigaglia, Conceptualization, Data curation, Formal analysis, Investigation, Methodology, Writing - original draft, Writing - review and editing; Sophie Peron, Data curation, Formal analysis, Funding acquisition, Investigation, Methodology, Writing - review and editing; Jeanne Eichelbrenner, Validation, Investigation; Sandra Chevalier, Julia Steger, Investigation, Methodology; Carine Barreau, Investigation, Methodology, Writing - review and editing; Evelyn Houliston, Conceptualization, Formal analysis, Writing - review and editing; Lucas Leclère, Conceptualization, Data curation, Formal analysis, Supervision, Funding acquisition, Validation, Investigation, Writing - original draft, Project administration, Writing - review and editing

### Author ORCIDs

Chiara Sinigaglia (iD) https://orcid.org/0000-0002-7195-7091
Evelyn Houliston (iD) http://orcid.org/0000-0001-9264-2585
Lucas Leclère (iD) https://orcid.org/0000-0002-7440-0467

### Decision letter and Author response

Decision letter https://doi.org/10.7554/eLife.54868.sa1
Author response https://doi.org/10.7554/eLife.54868.sa2

## Additional files

### Supplementary files
• Transparent reporting form

### Data availability

Transcriptomic data are deposited in ENA (EBI) - accession code PRJEB37920. All other data generated or analysed during this study are included in the manuscript and supporting files. Source data files have been provided.

The following dataset was generated:

| Author(s) | Year | Dataset title | Dataset URL | Database and Identifier |
|---|---|---|---|---|
| Peron S, Leclère L | 2020 | RNA-seq of *Clytia* umbrella fragments collected at different times, following manubrium ablation | https://www.ebi.ac.uk/ena/data/view/PRJEB37920 | ENA, PRJEB37920 |

The following previously published dataset was used:

| Author(s) | Year | Dataset title | Dataset URL | Database and Identifier |
|---|---|---|---|---|
| Leclère L, Copley R | 2018 | *Clytia hemisphaerica* reference transcriptome | https://www.ebi.ac.uk/ena/data/view/PRJEB28006 | ENA, PRJEB28006 |

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
