## [Decision Letter]

**Acceptance summary:**

This work provides a systematic analysis of regeneration in the medusa of the cnidarian Clytia hemisphaerica. Using a combination of amputations, tissue grafts, gene expression analyses, and pharmacological manipulations, the authors provide a beautiful description of regeneration in this organism, and how local interactions can shape global patterning. This work fills a significant gap in the field of comparative evolutionary studies of regeneration and provides a strong foundation for future work on this organism.

**Decision letter after peer review:**

Thank you for submitting your article "Pattern regulation in a regenerating jellyfish"; for consideration by *eLife*. Your article has been reviewed by three peer reviewers, one of whom is a member of our Board of Reviewing Editors, and the evaluation has been overseen by Marianne Bronner as the Senior Editor. The reviewers have opted to remain anonymous.

The reviewers have discussed the reviews with one another and the Reviewing Editor has drafted this decision to help you prepare a revised submission.

We would like to draw your attention to changes in our revision policy that we have made in response to COVID-19 (https://elifesciences.org/articles/57162). Specifically, when editors judge that a submitted work as a whole belongs in *eLife* but that some conclusions may require a modest amount of additional new data, as they do with your paper, we are asking that the manuscript be revised to either limit claims to those supported by data in hand, or to explicitly state that the relevant conclusions require additional supporting data.

Summary:

Sinigaglia and colleagues describe the processes underlying regeneration in medusae of the cnidarian *Clytia hemisphaerica*. Understanding the molecular mechanisms by which cnidarians regenerate is essential because they hold a key position in animal phylogeny as the sister taxa to bilaterians. Such knowledge this will help us to distinguish deeply conserved aspects of regenerative processes from taxon-specific inventions. Medusozoan cnidarians have two major life stages: polyp and medusa. Whereas *Hydra* and *Hydractinia* are well established for studying regeneration in polyps, very little is understood about medusa regeneration. This work extends the modern study of regeneration to the medusa forms of cnidarians. Using dissections, grafts, microscopy, pharmacological manipulations, and RNA-seq data, the authors (a) document events involved in wound closure, tissue remodeling, and establishment of signaling centres, and (b) demonstrate essential roles for mechanical forces, existing cells/tissues, and *wnt*/β-catenin signaling in these processes. This work fills a significant gap in the field of comparative/evolutionary studies of regenerative potential and builds a solid foundation for further research on this tractable model species. Overall, the experiments are well designed, and the findings are clearly reported. With some adjustments to clarify or revise conclusions about the interplay of mechanical factors and Wnt signaling as well as to appropriately acknowledge previous work, the paper will be suitable for publication in *eLife*.

Essential revisions:

1) One of the major conclusions of the paper is that mechanical forces activate the expression of Wnts and initiate Wnt signaling. (e.g. Introduction, last paragraph; Results subsection “Wnt signaling links muscle hub stabilization to manubrium regeneration”, last paragraph; the Discussion subsection “Mechanical control of regeneration in Clytia”). Perhaps, but the actual evidence to support this conclusion in the paper is not entirely convincing. The key observation is that expression of *wnt6* in dissected fragments was blocked by treatment with blebbistatin "in both S and L fragments (Figure 10E)." While this observation may hold for S fragments, Figure 10E shows no obvious expression of *wnt6* in control L fragments at 24 hpd, which is the only time-point shown for blebbistatin-treated fragments. Are there images that indicate lack of *wnt6* in L fragments at 8 hpd – a time when it would normally be expressed? If so, they would provide much stronger support for the causal link between mechanical forces and *wnt* expression. Perhaps more importantly, the observation that inhibition of Wnt signaling with PKF118-310 delays wound closure suggests that Wnt/β-catenin signaling may be upstream of mechanical forces, or at least interdependent of them. If stronger evidence to support induction of *wnts* by mechanical forces is not available, then the assertions mentioned above should be toned down.

2) The experiments described in the section "Why does only one manubrium usually regenerate?" overlap significantly with those previously conducted by Schmid et al., 1976 (note that the name of the journal in which this article appeared was not Development, Genes and Evolution, at the time of publication). However, only a general reference is made to this article in the Introduction of the current manuscript. This cursory mention, though respectfully made, is insufficient.

As in the current study, Schmid and co-workers found that an implanted manubrium could inhibit the regeneration of a new manubrium in bisected specimens, but only when the implant was close to the site at which the new manubrium would regenerate (Figure 3, Table 2). They likewise demonstrated that inhibition could not be caused by a diffusible factor or morphogen gradient (Figures 2 and 3, Tables 1 and 2). Furthermore, Schmid et al. contrasted this (physically mediated) inhibition with the (chemically induced) effect of the hypostome on regeneration in *Hydra* polyps (see p. 51, esp. second paragraph). The principle inhibitory mechanism proposed in the earlier study was competition for underlying tissues: a different model than the current study (402-427; Figure 11). However, Schmid et al. did discuss the role of tensile forces in regulating manubrium regeneration.

It is not acceptable for the authors of the current work to leave such earlier studies un-discussed. To demonstrate that their work offers new fundamental biological insight, they simply must do more to acknowledge the specific contributions of prior workers, and to discuss how their observations and models confirm, contradict, and/or extend the theoretical framework already available in the literature. We have not closely examined other historical papers for areas of overlap, but we presume that the authors will do so, and discuss relevant matters as appropriate in their revised manuscript.

3) Please clarify what defines a "hub," for readers' untrained eyes (subsection “Why does only one manubrium usually regenerate?”, second paragraph). The phalloidin-stained images in Figure 7H do have a rough focal point of phalloidin staining, albeit ring-shaped rather than with a central peak of intensity. However, when a bona fide manubrium regenerates, as in Figure 7F (also Figure 6C bottom right), central phalloidin staining naturally decreases as the manubrium extends and develops its lumen. Importantly, Figure 7H shows a 4 dpd specimen rather than an earlier time point; is there evidence that there was not a hub present earlier, as in Figure 6—figure supplement 1B at 24 hpd? Please provide more rationale for the conclusion that 7H shows a "non-hub," rather than slowed or arrested development after establishment of a stable hub. This is a key point to establish that the effect of a pre-existing (transplanted) manubrium is the same in manubrium-ablation experiments as in bisection experiments (e.g. L fragments in Figure 9). A time-series of staining would demonstrate whether a hub forms initially and then disaggregates, as in the bisection experiments, and the overall model proposed in Figure 11.

4) Subsection “Restoration of medusa form involves both body remodeling and regrowth of organs”, first paragraph: The data in Figure 1—figure supplement 2E showing manubrium regeneration does not appear to distinguish between fed and unfed animals, so it is unclear what supports the statement in the text that manubrium regeneration is stereotypical regardless of feeding levels.

5) The first paragraph of the subsection “Wound closure and body remodeling precede proliferation-dependent organ regeneration” presents a staging scheme to describe manubrium regeneration, which is an important framework for future studies. However, better drawings or photos are required to better illustrate this staging scheme for the reader. The text refers to Figure 1E, but not all stages are shown there (and they aren't labeled with stage numbers). Including such a figure is really critical when trying to establish a new staging scheme; it should not be left to the reader's imagination. Furthermore, it would be helpful to label images of regenerating manubria with this new stage classification throughout the manuscript.

6) Figure 2B, C: What is being compared in each statistical test? The legend needs to clarify this. The way the figure is drawn, it looks like each time point is being compared to the previous time point. It might make more sense to compare each time point to the baseline cell proliferation in an uninjured animal.

7) Figure 2—figure supplement 1B: Are the starved animals also regenerating? If yes, this is confusing since the text states that starved animals also regenerate a manubrium. If no, then it is unclear why the levels change over time. Either way, this figure needs clarification and further explanation.

8) Figure 10: How many of the presented in situ experiments were done and are the presented results representative? This information should be included. qPCR data would be nice to quantify these results, especially for the inhibitor treatments, but I understand it may not be possible to get back into the lab to do this right now.

9) Overall, this manuscript presents large amount of data and information, making it somewhat difficult for the reader to hold a coherent picture in their head. The model at the end certainly helped. Something that might help further: on the model figure one could include the figure panels that support each aspect of the model. That would help the reader to easily go back and assess to the underlying data for each aspect of the model.

---

## [Author Response]

Essential revisions:1) One of the major conclusions of the paper is that mechanical forces activate the expression of Wnts and initiate Wnt signaling. (e.g. Introduction, last paragraph; Results subsection “Wnt signaling links muscle hub stabilization to manubrium regeneration”, last paragraph; the Discussion subsection “Mechanical control of regeneration in Clytia”). Perhaps, but the actual evidence to support this conclusion in the paper is not entirely convincing. The key observation is that expression of wnt6 in dissected fragments was blocked by treatment with blebbistatin "in both S and L fragments (Figure 10E)." While this observation may hold for S fragments, Figure 10E shows no obvious expression of wnt6 in control L fragments at 24 hpd, which is the only time-point shown for blebbistatin-treated fragments. Are there images that indicate lack of wnt6 in L fragments at 8 hpd – a time when it would normally be expressed? If so, they would provide much stronger support for the causal link between mechanical forces and wnt expression. Perhaps more importantly, the observation that inhibition of Wnt signaling with PKF118-310 delays wound closure suggests that Wnt/β-catenin signaling may be upstream of mechanical forces, or at least interdependent of them. If stronger evidence to support induction of wnts by mechanical forces is not available, then the assertions mentioned above should be toned down.

In the revised manuscripts we have included data from additional experiments, including 8 hpd blebbistatin-treated S/L fragments assessed by in situ hybridisation and qPCR (new Figure 11). At 8 hpd, *CheWnt6* is expressed at the site of remodelling in both S and L fragments: *CheWnt6* expression is blocked by treatment with blebbistatin (8 hpd and 24 hpd), and can be re-activated by washing out the inhibitor. *CheWnt6* expression is retained only in fragments where the muscle ‘hub’ is stabilized. Together with the data presented previously, these results provide strong evidence for crosstalk between remodelling and *CheWnt6* expression, which we have tried to lay out clearly in the text. We agree that our data does not explain how *CheWnt6* is activated in the first place and have toned down accordingly the conclusion that *CheWnt6* is directly activated by mechanical forces.

2) The experiments described in the section "Why does only one manubrium usually regenerate?" overlap significantly with those previously conducted by Schmid et al. (1976 – note that the name of the journal in which this article appeared was not Development, Genes and Evolution, at the time of publication). However, only a general reference is made to this article in the Introduction of the current manuscript. This cursory mention, though respectfully made, is insufficient.As in the current study, Schmid and co-workers found that an implanted manubrium could inhibit the regeneration of a new manubrium in bisected specimens, but only when the implant was close to the site at which the new manubrium would regenerate (Figure 3, Table 2). They likewise demonstrated that inhibition could not be caused by a diffusible factor or morphogen gradient (Figures 2 and 3, Tables 1 and 2). Furthermore, Schmid et al. contrasted this (physically mediated) inhibition with the (chemically induced) effect of the hypostome on regeneration in Hydra polyps (see p. 51, esp. second paragraph). The principle inhibitory mechanism proposed in the earlier study was competition for underlying tissues: a different model than the current study (402-427; Figure 11). However, Schmid et al. did discuss the role of tensile forces in regulating manubrium regeneration.It is not acceptable for the authors of the current work to leave such earlier studies un-discussed. To demonstrate that their work offers new fundamental biological insight, they simply must do more to acknowledge the specific contributions of prior workers, and to discuss how their observations and models confirm, contradict, and/or extend the theoretical framework already available in the literature. We have not closely examined other historical papers for areas of overlap, but we presume that the authors will do so, and discuss relevant matters as appropriate in their revised manuscript.

We agree entirely that the extensive work of Schmid and colleagues is important to this study and warrants in-depth consideration. We have re-written parts of the Discussion and Results, to fully integrate these previous studies, citing and comparing the grafting results in particular.

3) Please clarify what defines a "hub," for readers' untrained eyes (subsection “Why does only one manubrium usually regenerate?”, second paragraph). The phalloidin-stained images in Figure 7H do have a rough focal point of phalloidin staining, albeit ring-shaped rather than with a central peak of intensity. However, when a bona fide manubrium regenerates, as in Figure 7F (also Figure 6C bottom right), central phalloidin staining naturally decreases as the manubrium extends and develops its lumen. Importantly, Figure 7H shows a 4 dpd specimen rather than an earlier time point; is there evidence that there was not a hub present earlier, as in Figure 6—figure supplement 1B at 24 hpd? Please provide more rationale for the conclusion that 7H shows a "non-hub," rather than slowed or arrested development after establishment of a stable hub. This is a key point to establish that the effect of a pre-existing (transplanted) manubrium is the same in manubrium-ablation experiments as in bisection experiments (e.g. L fragments in Figure 9). A time-series of staining would demonstrate whether a hub forms initially and then disaggregates, as in the bisection experiments, and the overall model proposed in Figure 11.

We define a “hub” as the convergence site of well-organized radial smooth muscle fibers. We added a new Figure 9—figure supplement 1 showing that indeed a hub forms in proximally-grafted animals (12 hpd) and then disorganizes, so that at later stages (48 hpd) the smooth muscle fibers do not meet at the wound site, but tend to converge towards the grafted manubrium. The principles governing hub assembly/disassembly are therefore comparable in the graft paradigm and in the S/L dissection. Given that the experimental variability between grafted specimens is much higher than between S/L fragments, it is not feasible to illustrate the time-course of events following manubrium grafting with fixed specimens reliably. We plan in the future to follow and analyze hub dynamics in individual grafted specimens by live imaging, for instance using transgenic muscle-reporter animals.

4) Subsection “Restoration of medusa form involves both body remodeling and regrowth of organs”, first paragraph: The data in Figure 1—figure supplement 2E showing manubrium regeneration does not appear to distinguish between fed and unfed animals, so it is unclear what supports the statement in the text that manubrium regeneration is stereotypical regardless of feeding levels.

We have not tested the efficiency of manubrium regeneration in relation to the previous feeding regime of the animals. We have now clarified this aspect in the Results.

5) The first paragraph of the subsection “Wound closure and body remodeling precede proliferation-dependent organ regeneration” presents a staging scheme to describe manubrium regeneration, which is an important framework for future studies. However, better drawings or photos are required to better illustrate this staging scheme for the reader. The text refers to Figure 1E, but not all stages are shown there (and they aren't labeled with stage numbers). Including such a figure is really critical when trying to establish a new staging scheme; it should not be left to the reader's imagination. Furthermore, it would be helpful to label images of regenerating manubria with this new stage classification throughout the manuscript.

References to our staging system for manubrium regeneration have been added where appropriate (Figures 1, 2, 3, 10, 12). One additional time-point (12 hpd) has been added to Figure 1E. We also added drawing in Figure 2A to illustrate the different stages of manubrium regeneration.

6) Figure 2B, C: What is being compared in each statistical test? The legend needs to clarify this. The way the figure is drawn, it looks like each time point is being compared to the previous time point. It might make more sense to compare each time point to the baseline cell proliferation in an uninjured animal.

Each test compared a time-point and the previous one, in order to assess the evolution of EdU-positive cells numbers during the course of regeneration. We did not think that a comparison with the proliferation levels in a non-dissected manubrium would be informative, because the endogenous manubrium is much bigger and contains very different stem cell populations, for instance supporting continuous proliferation of nematocyte progenitors. The legend has now been modified accordingly.

7) Figure 2—figure supplement 1B: Are the starved animals also regenerating? If yes, this is confusing since the text states that starved animals also regenerate a manubrium. If no, then it is unclear why the levels change over time. Either way, this figure needs clarification and further explanation.

In the experiment presented in Figure 2—figure supplement 1B, ‘’starved’’ animals were not dissected (and thus were not regenerating). Manubrium-dissected jellyfish cannot feed, hence the most appropriate comparison of proliferation levels during regeneration is to animals that have been starved for an equivalent amount of time. We have clarified the labels in the figure.

8) Figure 10: How many of the presented in situ experiments were done and are the presented results representative? This information should be included. qPCR data would be nice to quantify these results, especially for the inhibitor treatments, but I understand it may not be possible to get back into the lab to do this right now.

The in situ hybridization experiments were performed at least twice for each condition, and gave highly reproducible results; this information was added to the Materials and methods section. Sample sizes have now been added to the figures (Figures 10 and 11). We agree that qPCR can provide a quantitative picture of gene expression changes, and so have now included these additional experiments to Figures 10, 11, and Figure 11—figure supplement 1. The qPCR results confirmed the in situ hybridization data, but also showed an unexpected effect of blebbistatin on the expression of *CheWnt6* across all umbrella tissues unrelated to remodelling or regeneration. We thus compared *CheWnt6* expression close to the remodelling site (“f”) to the expression in fragment of the surrounding umbrella tissues (“F”) of comparable size (Figures 11C and Figure 11—figure supplement 1). The ratio of expression levels (relative to housekeeping genes) showed a marked increase of *CheWnt6* levels at the wound site of control-treated remodelling jellyfish, increment that was not detected in blebbistatin-treated animals.

9) Overall, this manuscript presents large amount of data and information, making it somewhat difficult for the reader to hold a coherent picture in their head. The model at the end certainly helped. Something that might help further: on the model figure one could include the figure panels that support each aspect of the model. That would help the reader to easily go back and assess to the underlying data for each aspect of the model.

We have now added to the model (Figure 12) references to the corresponding figures in the manuscript.